# Direct measurement of TRPV4 and PIEZO1 activity reveals multiple mechanotransduction pathways in chondrocytes

M Rocio Servin-Vences[1], Mirko Moroni[1], Gary R Lewin[1]*, Kate Poole[1,2,3]*

[1]Department of Neuroscience, Max Delbruck Center for Molecular Medicine, Berlin, Germany; [2]Department of Physiology, School of Medical Sciences, University of New South Wales, Sydney, Australia; [3]EMBL Australia node for Single Molecule Sciences, School of Medical Sciences, University of New South Wales, Sydney, Australia

**Abstract** The joints of mammals are lined with cartilage, comprised of individual chondrocytes embedded in a specialized extracellular matrix. Chondrocytes experience a complex mechanical environment and respond to changing mechanical loads in order to maintain cartilage homeostasis. It has been proposed that mechanically gated ion channels are of functional importance in chondrocyte mechanotransduction; however, direct evidence of mechanical current activation in these cells has been lacking. We have used high-speed pressure clamp and elastomeric pillar arrays to apply distinct mechanical stimuli to primary murine chondrocytes, stretch of the membrane and deflection of cell-substrate contacts points, respectively. Both TRPV4 and PIEZO1 channels contribute to currents activated by stimuli applied at cell-substrate contacts but only PIEZO1 mediates stretch-activated currents. These data demonstrate that there are separate, but overlapping, mechanoelectrical transduction pathways in chondrocytes.

*For correspondence: glewin@ mdc-berlin.de (GRL); k.poole@ unsw.edu.au (KP)

**Competing interests:** The authors declare that no competing interests exist.

## Introduction

In diarthrodial joints, which allow a large degree of movement, the surfaces of the opposing bones are lined with hyaline cartilage which reduces friction. This tissue is avascular and non-innervated and comprised of individual chondrocytes embedded in an extracellular matrix (ECM). Production and homeostatic maintenance of cartilage structure is dependent on chondrocytes (*Hall et al., 1996*). Chondrocytes sense changes in the physical microenvironment and mechanical loading within the joints and adjust the balance of anabolic and catabolic processes to maintain the integrity and physical properties of the ECM (*Buckwalter and Mankin, 1997a*; *Goldring and Marcu, 2009*). Disrupting these homeostatic processes can lead to osteoarthritis (OA) whereby inappropriate activation of catabolic pathways leads to cartilage degradation (*Buckwalter and Mankin, 1997b*). It is therefore important to define how chondrocytes respond to mechanical stimuli and to understand how the sensitivity of the mechanotransduction pathways is modulated as both excessive and insufficient mechanical loading of the joint can lead to joint dysfunction.

Chondrocytes are embedded within a complex, viscoelastic environment formed by specialized ECM, proteoglycans and water (*Sophia Fox et al., 2009*; *Mow et al., 1984*). Physiologically, the cartilage is subjected to a spectrum of mechanical inputs (*Sanchez-Adams and Athanasiou, 2011*). Cartilage is regularly impacted by compressive forces that are initially carried by the fluid phase, before being transferred to the elastic ECM molecules within the tissue (*Mow et al., 1980*). The movement

**eLife digest** Cartilage is a flexible tissue that cushions the joints in our body, allowing them to move smoothly. It is made of cells called chondrocytes that are surrounded by a scaffold of proteins known as the extracellular matrix. Chondrocytes regularly experience mechanical forces, which can arise from the movement of fluid within the joints or be transmitted to chondrocytes via the extracellular matrix. These cells sense mechanical forces by a process known as mechanotransduction, which allows chondrocytes to alter the composition of the extracellular matrix in order to maintain an appropriate amount of cartilage. If mechanotransduction pathways are disrupted, the cartilage may become damaged, which can result in osteoarthritis and other painful joint diseases.

The membrane that surrounds a chondrocyte contains proteins known as ion channels that are responsible for sensing mechanical forces. The channels open in response to mechanical forces to allow ions to flow into the cell. This movement of ions generates electrical signals that result in changes to the production of extracellular matrix proteins. However, there is little direct evidence that mechanical forces can activate ion channels in chondrocytes and it not known how these cells respond to different types of forces.

To address these questions, Servin-Vences et al. exposed chondrocytes from mice to mechanical forces either at the point of contact between the cell and its surrounding matrix, or to stretch the cell membrane. The experiments show that two ion channels called PIEZO1 and TRPV4 both generate electrical currents in response to forces transmitted between cells and the extracellular matrix. However, only PIEZO1 generates a current when the cell membrane is stretched. Thus, chondrocytes are able to distinguish between different types of mechanical forces.

More work is needed to understand how mechanical forces are able to activate these ion channels. Understanding how these processes work at the molecular level will hopefully lead to new therapies that boost cartilage production to treat joint diseases.

of fluid within the joints also generates shear forces (*Wong et al., 2008*), whereas tensile forces are transmitted to chondrocytes via the surrounding pericellular matrix (PCM) (*Guilak et al., 2006*). Given the biomechanical complexity of this system, it is difficult to model precisely how these various mechanical inputs are experienced by the cells; however, in the simplest terms, cells will experience mechanical stimuli propagated both via the fluid phase and via the matrix to which the cells are bound.

Cellular mechanotransduction depends on a number of distinct processes (*Roca-Cusachs et al., 2012*) including channel-mediated ionic flux across the membrane (*Nilius and Honoré, 2012*; *Martinac, 2004*; *Arnadóttir and Chalfie, 2010*), integrin-mediated signaling (*Chen et al., 2004*; *Schwartz, 2010*), action of strain gauge proteins (*Hirata et al., 2008*) or cytoskeleton-mediated transfer of mechanical signals from the plasma membrane to the nucleus (*Maniotis et al., 1997*). In chondrocytes, a number of these pathways have been implicated in the mechanotransduction that is required for homeostasis; however, in this study, we focus on the role of mechanically gated ion channels. We refer here to channel-mediated mechanotransduction as *mechanoelectrical transduction* in order to distinguish this process from parallel mechanotransduction mechanisms.

It has long been proposed that ion channels play a role in the process of chondrocyte mechanotransduction. Hyperpolarization of chondrocytes on application of mechanical loads is inhibited (*Wright et al., 1996*) and matrix production is altered (*Mouw et al., 2007*) in the presence of GdCl$_3$, a non-specific inhibitor of mechanically gated ion channel activity. Blocking the TRPV4 ion channel using a specific antagonist (GSK205) inhibits matrix production in response to compressive mechanical stimulation and the TRPV4 agonist, GSK1016790A, stimulates matrix production in the absence of mechanical stimulation (*O'Conor et al., 2014*). Additionally, mutations in the human *TRPV4* gene can lead to joint dysfunction (*Lamandé et al., 2011*; *Loukin et al., 2010*). In mouse models, a global *Trpv4*$^{-/-}$ knockout leads to an increased susceptibility to obesity-induced (*O'Conor et al., 2013*) and age-related OA (*Clark et al., 2010*), whereas conditional knockout of *Trpv4* in adult cartilage decreases the risk of age-related OA (*O'Conor et al., 2016*). Despite this

growing body of evidence that TRPV4 is directly involved in chondrocyte mechanotransduction, no evidence for gating of TRPV4 by mechanical stimuli (other than osmotic stimuli (*Lechner et al., 2011*), and membrane-stretch in *Xenopus laevis* oocytes (*Loukin et al., 2010*)) has been presented. More recently, it has been shown that $Ca^{2+}$ spikes in isolated porcine chondrocytes (detected using $Ca^{2+}$ imaging) are reduced when the mechanically gated *Piezo1* and *Piezo2* channel transcripts are knocked down using siRNA (*Lee, 2014*).

Both PIEZO1 and PIEZO2 have been demonstrated to mediate mechanically gated ion currents in neuronal cells and neuronal cell lines (*Coste et al., 2012*; *Ranade et al., 2014a*). Beyond the nervous system, PIEZO1 has been found to be functionally relevant in the vasculature (*Li et al., 2014*; *Ranade et al., 2014b*), urothelium (*Miyamoto et al., 2014*), tubal epithelial cells (*Peyronnet et al., 2013*), erythrocytes (*Zarychanski et al., 2012*), as well as in porcine chondrocytes (*Lee, 2014*). However, in these non-neuronal cell types there has, to date, only been one publication that has directly measured mechanical activation of ion channels in intact cells and a reduction in channel gating when PIEZO1 is absent (*Peyronnet et al., 2013*). What has been lacking is: (1) a direct demonstration of mechanically gated channel activity in chondrocytes; (2) a quantitative analysis of the relative contributions of distinct mechanically gated ion channels in chondrocyte mechanotransduction and (3) an analysis of how chondrocytes respond to distinct mechanical stimuli.

Here, we have used an experimental approach wherein we apply mechanical stimuli at cell-substrate contact points and concurrently monitor membrane currents using whole-cell patch-clamp (*Poole et al., 2014*). This approach allows us to measure channel activity in response to mechanical stimuli that are applied via connections to the substrate. Using this approach, we show that we can measure mechanically gated currents in intact chondrocytes. To the best of our knowledge, these measurements represent the first direct demonstration of mechanically gated ion channel activity in primary chondrocytes. We have further demonstrated that both the TRPV4 and PIEZO1 channels contribute to this current and that, in particular for TRPV4, the nature of the membrane environment and applied stimulus are crucial for channel gating.

## Results

### Primary, murine chondrocyte cultures

To study mechanically gated ion channels in chondrocytes, we prepared primary cells from mouse articular cartilage isolated from the knees and femoral heads of 4- to 5-day-old mouse pups. A fraction of these cells were encapsulated in alginate beads and the remainder seeded in 2D tissue culture flasks. The chondrocytes cultured in alginate beads retained the chondrocyte phenotype (high levels of *Sox9* transcript, spherical morphology and staining for SOX9 and Collagen X [*Lefebvre et al., 1997, 2001*; *Dy et al., 2012*; *Poole et al., 1984*; *Ma et al., 2013*]) (*Figure 1A–B*). The cells seeded in tissue culture flasks dedifferentiated away from the chondrocyte phenotype, as reflected in reduced levels of *Sox9* transcript, a fibroblast-like morphology (*Caron et al., 2012*) and negative staining for SOX9 and Collagen X (*Figure 1B*). Dedifferentiated cells from tissue culture flasks were redifferentiated back into the chondrocyte phenotype by encapsulating them in alginate for 7 days (*Figure 1*, *Figure 1—figure supplement 1*). We found that SOX9-positive cells exhibited a spherical morphology and that the average diameter of these cells was $11.7 \pm 2.0\ \mu m$ (mean ± s.d., n = 77 cells) (*Figure 1—figure supplement 1*). Accordingly, the cells with a chondrocyte phenotype could be distinguished on the basis of their morphology and selected for study using bright-field microscopy in a live, 2D culture.

### Measuring mechanically gated ion channel activity at the cell-substrate interface

Within the cartilage, mechanical stimuli are transferred to chondrocytes via the surrounding PCM (*Guilak et al., 2006*). We tested whether the regions of the membrane that form the cell-substrate interface constitute an important compartment for mechanoelectrical transduction. We seeded chondrocytes on an elastomeric pillar array cast in polydimethylsiloxane (PDMS) where each element of the array had defined dimensions and each cell-substrate contact point was $10\ \mu m^2$ (*Figure 2A*) (*Poole et al., 2014*). A glass probe (driven by a Piezo-electric element) was used to

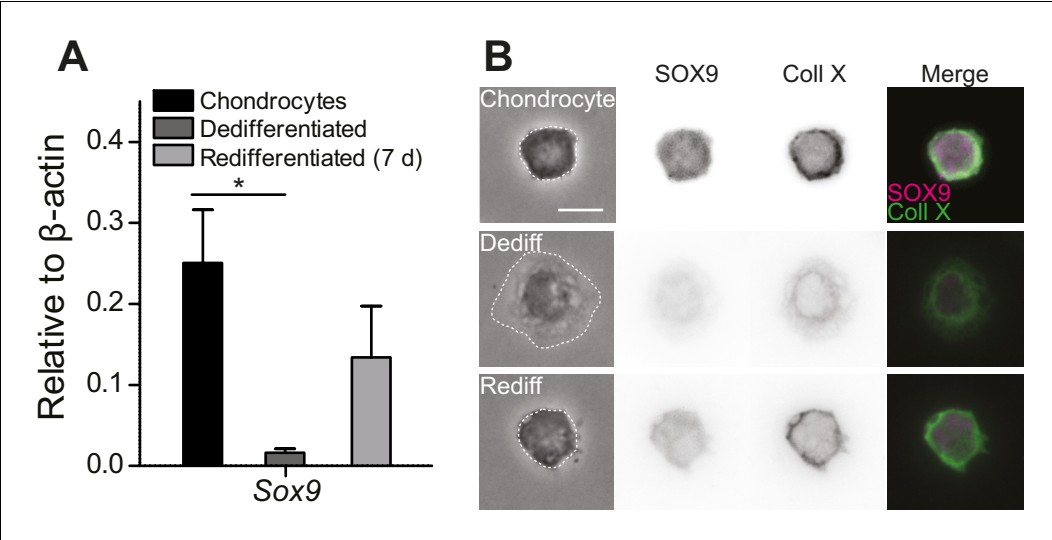

**Figure 1.** Primary, murine chondrocyte culture. (**A**) Transcript levels of the transcription factor *Sox9* in just harvested chondrocytes, dedifferentiated cells (post 7 days in monolayer culture) and redifferentiated chondrocytes (recovered from 2D plastic and encapsulated in alginate for 7 days). Data are displayed as mean ± s. e.m. Note, significantly less *Sox9* transcript was detected in the population of dedifferentiated cells (one-way ANOVA, Tukey Post-hoc test *p=0.035; n $\geq$ 3.) (**B**) Phase contrast and epi-fluorescent images representative of the morphological differences between chondrocytes, dedifferentiated and redifferentiated cells. SOX9 was detected in the nucleus and Collagen X at the membrane of chondrocytes and redifferentiated cells, but not the dedifferentiated population (inverted images and overlay). Scale bar 10 μm.

The following figure supplement is available for figure 1:

**Figure supplement 1.** Schematic diagram of the isolation and culture of primary murine chondrocytes.

deflect an individual pilus in order to apply a series of fine deflection stimuli to the cell directly at the cell-substrate interface (for range of deflections see *Figure 2A*).

In order to analyze chondrocyte mechanoelectrical transduction, cells were released from alginate and seeded over pillar arrays coated with poly-ι-lysine (PLL). The cells attached and initially exhibited the spherical morphology typical of chondrocytes. Within 3 hr, the morphology of a subset of cells became more fibroblast-like as the cells dedifferentiated. We investigated whether the chondrocytes and the cells that had dedifferentiated in situ exhibited similar mechanoelectrical transduction properties in order to determine if these cells with distinct morphologies could be treated as a coherent sample. The application of stimuli to the chondrocytes evoked deflection-gated inward currents in 88.9% of cells (*Figure 2B*) (24/27 cells). Deflection-gated currents were also observed in dedifferentiated cells (*Figure 2C*) (88.2% (15/17 cells)). The kinetics of these currents suggested a channel directly gated by mechanical stimuli (chondrocyte currents: latency = 3.6 ± 0.3 ms, activation time constant ($\tau_1$) = 1.7 ± 0.3 ms, dedifferentiated cell currents: latency = 3.1 ± 0.3 ms, $\tau_1$ = 1.4 ± 0.3 ms, mean ± s.e.m., n = 99 and 109 currents, measured across 24 chondrocytes and 15 dedifferentiated cells) (*Figure 2D*). We found that both the latency and the $\tau_1$ values were significantly faster for currents measured in the dedifferentiated cells (Mann-Whitney U test, p=0.018, p=0.04, respectively). In addition, whilst no significant difference was noted in the $\tau_2$ values (p=0.19), the variance in the $\tau_2$ of currents measured in dedifferentiated cells was significantly higher compared to chondrocytes (F test, p<0.0001, n = 109 and 99 currents, respectively). These data demonstrate ion channel-mediated mechanoelectrical transduction in chondrocytes. Such measurements have previously proven impossible due to application of techniques incompatible with simultaneous patch-clamp analysis or that result in the destruction of cellular integrity before any mechanical activation of ion channels can be observed, such as cellular indentation of chondrocytes (*Lee, 2014*).

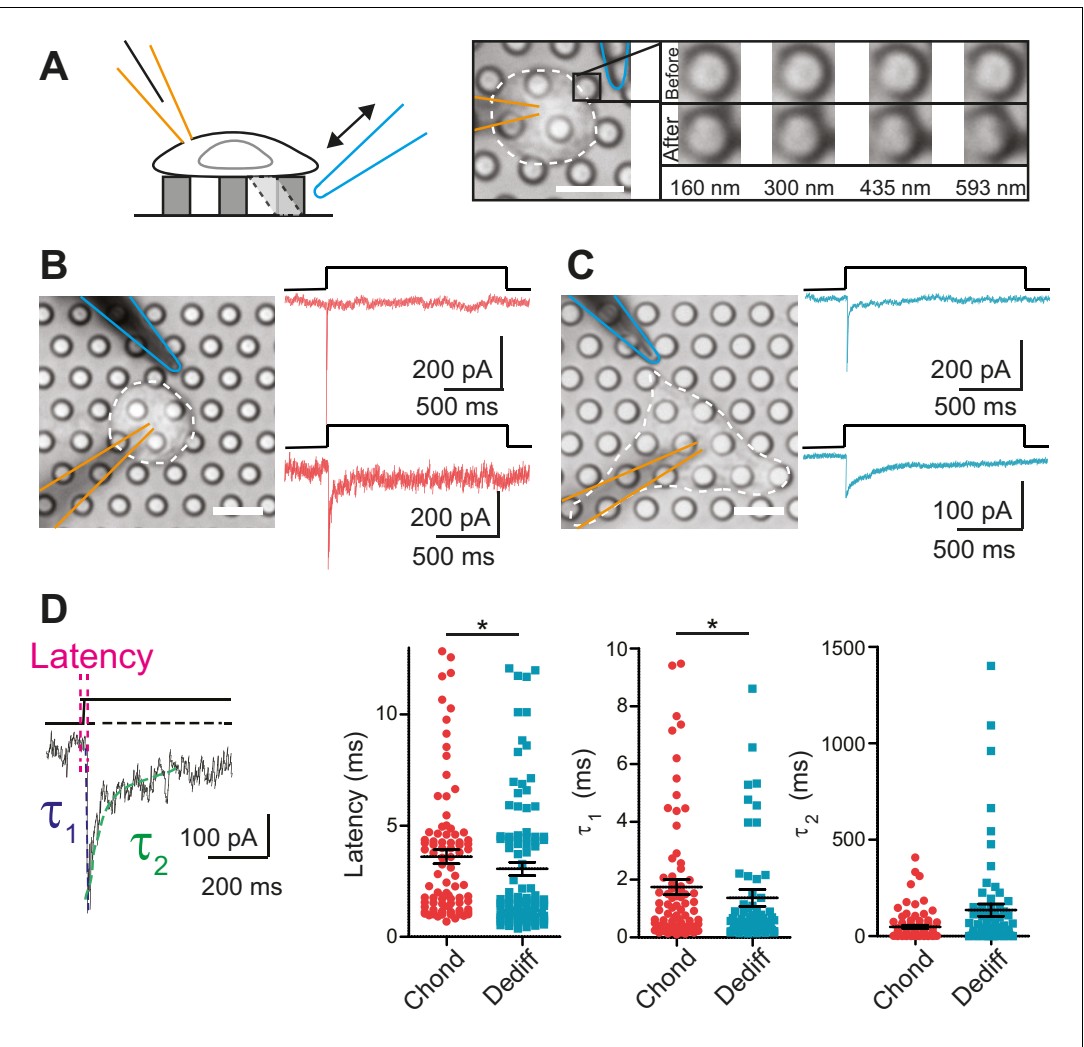

**Figure 2.** Mechanoelectrical transduction currents in primary cells isolated from mouse cartilage. (**A**) Deflection stimuli applied via cell-matrix contact points. Left panel: cartoon of pillar array experiment, stimuli are applied by deflecting a pilus subjacent to a cell that is concurrently monitored using whole-cell patch-clamp (blue indicates stimulator probe and orange the patch pipette.) Right panel: bright-field image of a chondrocyte seeded on the pillar array. Successive images of the movement of the highlighted pilus demonstrate the degree of movement corresponding to the stimuli used in this study (**B**) Deflection-gated mechanoelectrical transduction currents in chondrocytes. Bright-field image of a chondrocyte and corresponding example traces of deflection-gated currents (red). (**C**) Deflection-gated mechanoelectrical transduction currents in dedifferentiated cells. Bright-field image of a dedifferentiated cell and representative traces of deflection-gated currents (blue). (**D**) Comparison of current kinetics. Left panel indicates values measured (latency (magenta), activation time constant ($\tau_1$, blue) and current decay ($\tau_2$, green)). Data are displayed as individual values (chondrocytes: red, dedifferentiated cells: cyan), mean ± s.e.m. superimposed in black.

The following source data is available for figure 2:

**Source data 1.** Electrophysiological characteristics of WT chondrocytes and WT dedifferentiated cells.

## Chondrocytes and dedifferentiated cells display distinct mechanosensitivity

An advantage of applying stimuli via pillar arrays is that the stimuli are applied to a defined area of membrane. We therefore quantified the magnitude of each applied stimulus, and compared the sensitivity of mechanoelectrical transduction in distinct subsets of cells. Each individual pilus acts as a

light guide, such that the center can be calculated from a 2D Gaussian fit of intensity values within a bright-field image (*du Roure et al., 2005*). An image was taken before, during and after the stimulus, and the magnitude of each deflection was subsequently calculated from the difference between the coordinates of the center of the pilus in successive images.

In order to collect stimulus-response data, we applied stimuli across the range 1–1000 nm to each cell and measured the currents that were evoked. To compare the sensitivity of the mechanoelectrical transduction in chondrocytes versus dedifferentiated cells, our analysis included only those cells that responded to at least one stimulus within the 1–1000 nm range. We binned current amplitude data by stimulus size and averaged across cells for each bin (*Figure 3A*). We found that stimuli within the ranges of 10–50 nm and 250–500 nm produced significantly larger currents in the dedifferentiated cells, in comparison with chondrocytes (Mann-Whitney test, for the range 10 nm to 50 nm p=0.02 and for 100 nm to 250 nm p=0.004) (*Figure 3A*). When the stimulus-response data was compared using two-way ANOVA, the response of the chondrocytes was significantly different to that of the dedifferentiated cells (*Figure 3A*; 24 chondrocytes vs 15 dedifferentiated cells, p=0.03). In addition, the smallest stimulus required to gate currents was significantly lower for the dedifferentiated cells, compared to chondrocytes (59 ± 13 nm (mean ± s.e.m., 15 cells); 252 ± 68 nm (mean ± s.e.m., 24 cells), Mann-Whitney test p=0.028) (*Figure 3B*). We conclude that, compared to chondrocytes, the dedifferentiated cells were more sensitive to deflection stimuli applied at cell-substrate contact points.

Many cell-types exhibit stretch-activated currents when pressure-stimuli are applied to membrane patches (*Sachs, 2010*). Using high-speed pressure-clamp (HSPC) on outside-out patches, we detected stretch-activated currents in both chondrocytes and dedifferentiated cells (*Figure 3C*). Analysis of the $P_{50}$ showed that there was no significant difference between the sensitivity of stretch-activated currents in chondrocytes (87.1 ± 6.0 mmHg, mean ± s.e.m., n = 12) compared to dedifferentiated cells (78.7 ± 7.4 mmHg, mean ± s.e.m., n = 13) (*Figure 3D*). These data suggest that the pressure-generated mechanoelectrical transduction in membrane patches is a separable phenomenon from deflection-gated currents observed when stimuli are applied at cell-substrate contact points. Due to the significant differences in mechanoelectrical transduction in response to deflection stimuli in chondrocytes versus dedifferentiated cells all further experiments were conducted on the population of cells exhibiting the chondrocyte phenotype.

## Molecules of mechanotransduction expressed in chondrocytes

We used RT-qPCR analysis to determine if *Piezo1* and *Piezo2* transcript could be detected in murine chondrocytes and to confirm the presence of *Trpv4* transcript in these cells. We found significant levels of *Trpv4* and *Piezo1* transcript; however, *Piezo2* transcript could not be reliably detected in our samples, in contrast to the observations made for porcine chondrocytes (*Lee, 2014*) (*Figure 4—figure supplement 1*).

## Substrate-deflection sensitive currents in chondrocytes depend, in part, on both PIEZO1 and on TRPV4

In order to directly test whether the PIEZO1 channels are involved in chondrocyte mechanoelectrical transduction, we used validated miRNA constructs (*Poole et al., 2014*) to reduce PIEZO1 levels and examined the resulting effect on deflection-gated mechanoelectrical transduction currents. We transfected dedifferentiated cells with a plasmid encoding the *Piezo1*-targeting miRNA or a scrambled miRNA. Cells were recovered from culture flasks and redifferentiated in alginate beads, before harvesting and seeding onto pillar arrays. Cells expressing the GFP marker were selected for measurement. The percentage of cells that responded to stimuli within the 1–1000 nm range was significantly reduced when chondrocytes were treated with *Piezo1*-targeting miRNA (50%, 6/12 cells), in comparison with those cells treated with the scrambled miRNA (19/22 cells, Fisher's exact test, p=0.04) (*Figure 4A*). These data show that knocking down the levels of the PIEZO1 channel reduces the likelihood of evoking deflection-gated currents. When the stimulus-response data was plotted, the PIEZO1 knockdown cells showed a tendency for reduced mechanoelectrical transduction, compared to control cells (*Figure 4B*).

TRPV4 has been proposed to play a role in chondrocyte mechanoelectrical transduction (*Clark et al., 2010*; *Leddy et al., 2014*; *Dunn et al., 2013*). We therefore studied deflection-gated

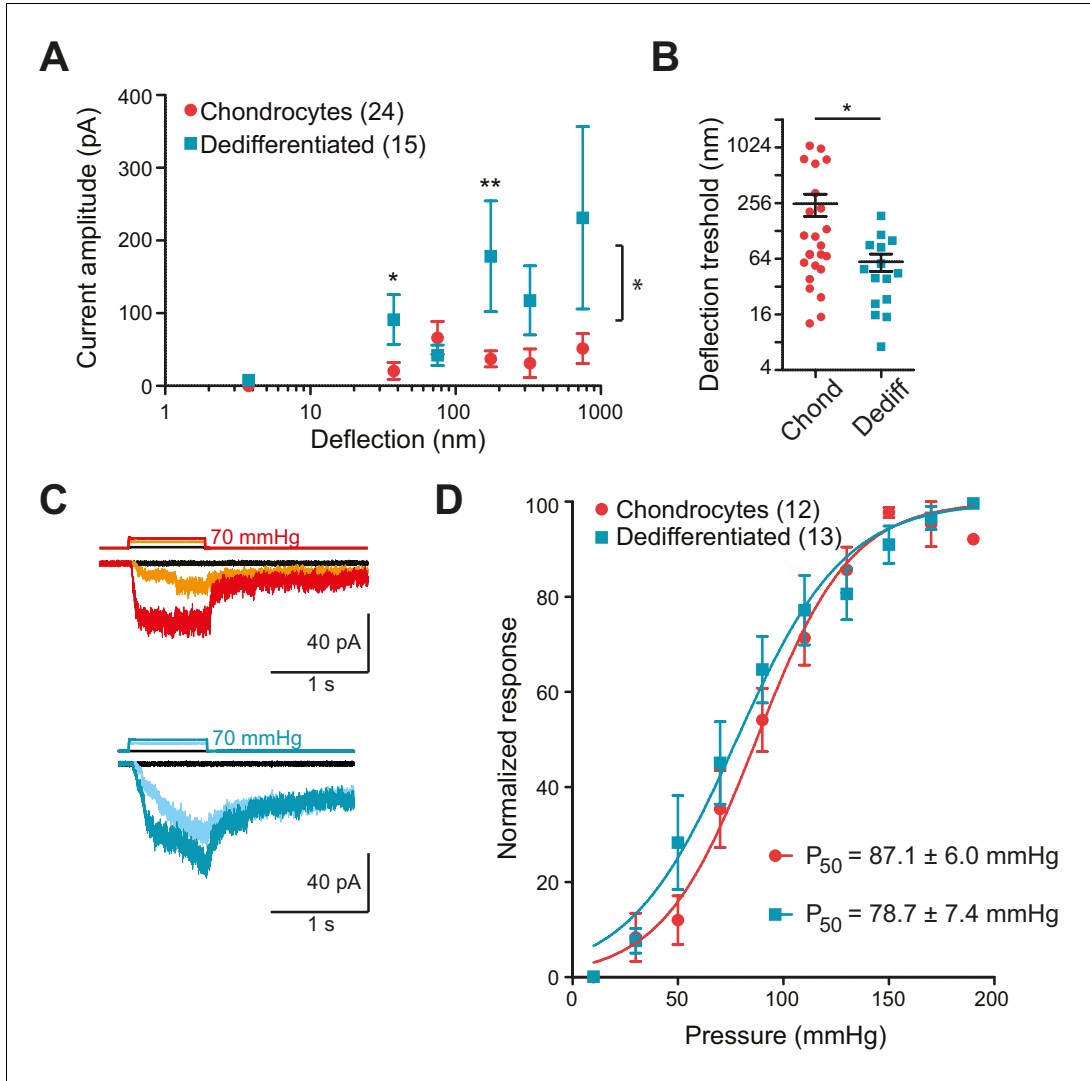

**Figure 3.** Chondrocytes and dedifferentiated cells display distinct mechanosenstivity to substrate deflections. (**A**) Stimulus-response graph of deflection-gated currents in chondrocytes (red circles) and dedifferentiated cells (cyan squares). Measurements from an individual cell were binned according to stimulus size and current amplitudes were averaged within each bin, then across cells, data are displayed as mean ± s.e.m. For stimuli between 10–50 and 100–250 nm, the dedifferentiated cells exhibit significantly larger currents. (Mann-Whitney test *p=0.02 and **p=0.004, respectively, n = 24 chondrocytes and 15 dedifferentiated cells.) Additionally, an ordinary two-way ANOVA indicates that the cell-types differ in their overall response (*p=0.03). (**B**) Chondrocytes and dedifferentiated cells display distinct deflection thresholds to substrate deflections. A threshold was calculated by averaging the smallest deflection that resulted in channel gating, for each cell. The threshold for chondrocytes, 252 ± 68 (mean ± s.e.m., n = 24) was significantly higher than that calculated for dedifferentiated cells 59 ± 13 (mean ± s.e.m., n = 15) (Mann-Whitney, *p=0.028). (**C**) Representative traces from HSPC recordings of stretch-activated currents from outside-out patches pulled from chondrocytes (upper panel) and dedifferentiated cells (lower panel). (**D**) Stimulus-response curve of pressure-gated currents in chondrocytes (red) and dedifferentiated cells (cyan), normalized to maximal amplitude measured for each sample. (Data are displayed as mean ± s.e.m., n = 12 chondrocytes, 13 dedifferentiated cells.).

The following source data is available for figure 3:

**Source data 1.** Statistical comparison of mechanoelectrical transduction currents, chondrocytes vs dedifferentiated cells.

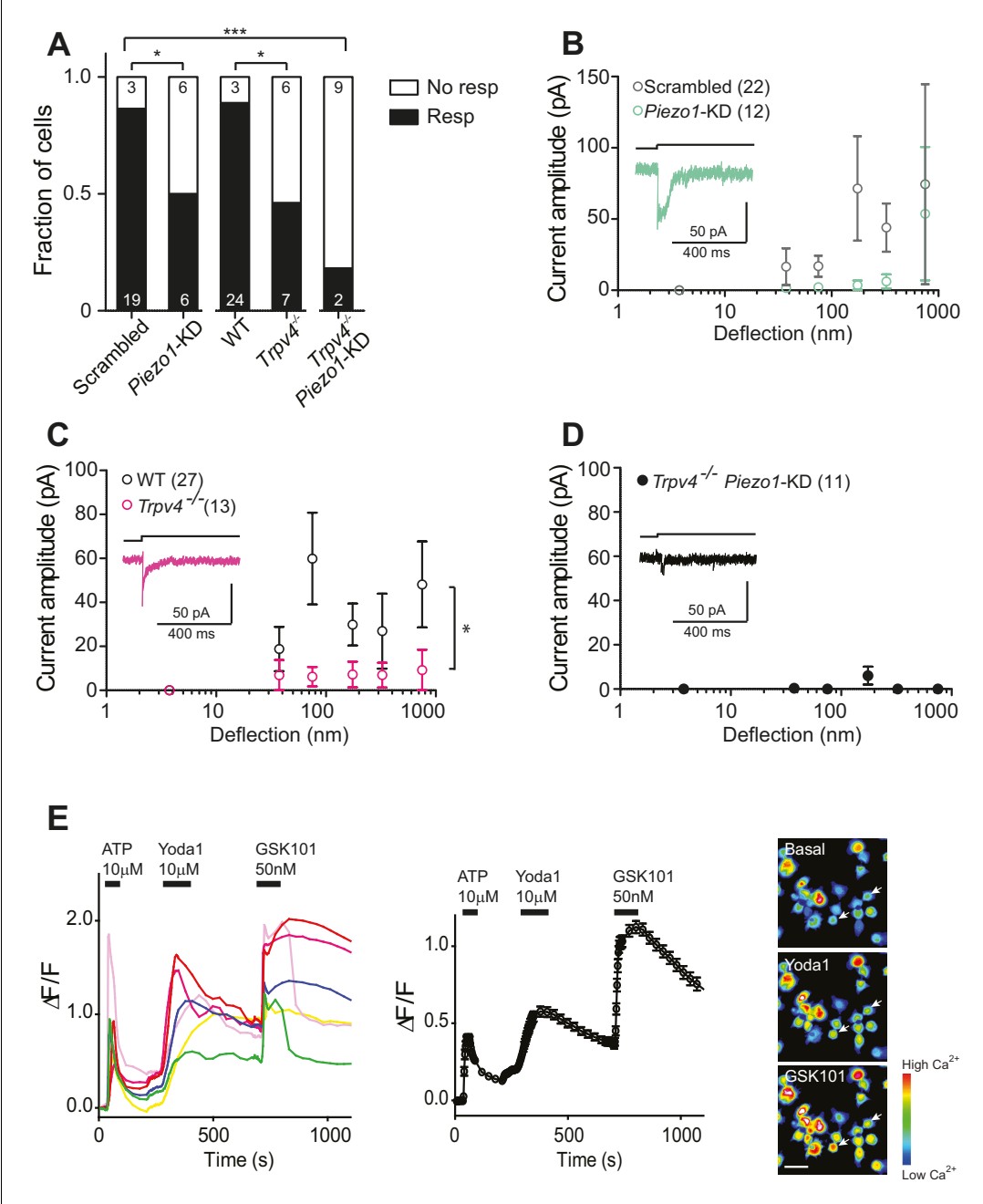

**Figure 4.** Substrate-deflection gated currents are mediated by PIEZO1 and TRPV4. (**A**) Fraction of chondrocytes that responded with at least with one mechanically gated current in response to deflection stimuli. Knockdown of *Piezo1* resulted in significantly fewer responsive cells compared with cells treated with non-targeting miRNA (scrambled) (Fisher´s exact test, *p=0.04). *Trpv4*[-/-] chondrocytes were significantly less likely to respond to deflection stimuli compared with WT cells (Fisher's exact test, *p=0.03). When the miRNA against *Piezo1* was expressed in *Trpv4*[-/-] chondrocytes, the response further decreased compared with the WT chondrocytes transfected with a scrambled miRNA (***p=0.002, Fisher's exact test). (**B**) Stimulus-response graph of the deflection-gated currents in chondrocytes transfected with a scrambled miRNA (gray open circles, n = 22 cells) or *Piezo1*-targeting miRNA (green open circles, n = 12 cells). Data are displayed as mean ± s.e.m., and a representative trace of the mechanosensitive currents is shown as insert (green line). (**C**) Cells isolated from a *Trpv4*[-/-] mouse are significantly less sensitive to deflections, in comparison with WT cells. Stimulus-response graph of the mechanically gated currents triggered by pillar deflections in WT chondrocytes (black open circles, n = 27 cells) and *Trpv4*[-/-] chondrocytes (magenta open circles, n = 13 cells). The *Trpv4*[-/-] cells are significantly less responsive to substrate deflections (ordinary two-way ANOVA, *p=0.04). Data are displayed as mean ± s.e.m. A representative trace is shown as insert (magenta line). (**D**) Stimulus-response graph of *Trpv4*[-/-] chondrocytes transfected with a *Piezo1*-targeting miRNA. Data are displayed as mean ± s.e.m. (n = 11 cells). Chondrocytes from the *Trpv4*[-/-] mouse treated with *Piezo1*-targeting miRNA were significantly less sensitive to substrate deflections, in comparison with WT cells treated with scrambled miRNA (ordinary

*Figure 4 continued on next page*

*Figure 4 continued*

two-way ANOVA, *p=0.04). A representative trace is shown as insert (black line). (E) Flourometric calcium imaging of chondrocyte responses to Yoda1 and GSK1016790A. Cells were perfused with ATP (10 μM), Yoda1(10 μM) and GSK1016790A (GSK101, 50 nM) as indicated by black bars and changes in $[Ca^{2+}]$ were monitored by using the $Ca^{2+}$ responsive dye, Cal520. In the left panel, traces correspond to intensity changes in individual cells and in the right panel is a plot representing the average of all cells (as mean ± s.e.m.). Example images are presented of cells before activation, during application of Yoda1 and of GSK1016790A. Scale bar 20 μm. Each cell that responded to ATP was included in the analysis (400 cells, two preparations).

The following source data and figure supplement are available for figure 4:

**Source data 1.** Electrophysiological characteristics of WT, *Trpv4*$^{-/-}$ and miRNA-treated chondrocytes.
**Figure supplement 1.** Normalized transcript levels of *Piezo1*, *Piezo2* and *Trpv4* in primary chondrocytes.

currents in chondrocytes isolated from *Trpv4*$^{-/-}$ mice (*Suzuki et al., 2003*) (back-crossed onto a C57Bl/6 background). Cells were isolated and cultured in the same fashion as wild-type (WT) cells. We found that deflection-gated currents could be observed in a subset of *Trpv4*$^{-/-}$ chondrocyte yet only 46.2% (6/13 cells) responded to deflections within the range of 1–1000 nm, significantly less than the percentage of responsive WT cells, 88.9% (24/27 cells) (Fisher's exact test, p=0.03) (*Figure 4A*). It was challenging to characterize the kinetics of the few, remaining currents. However, the latency between stimulus and channel gating was significantly longer in *Trpv4*$^{-/-}$ chondrocytes (7.8 ± 1.6 ms) compared with WT chondrocytes (3.6 ± 0.3 ms) (mean ± s.e.m., n = 12 and 99 currents, respectively, Mann-Whitney test, p=0.015). The stimulus-response plot was significantly different in WT chondrocytes vs *Trpv4*$^{-/-}$ chondrocytes (two-way ANOVA, p=0.04) (*Figure 4C*).

These data clearly indicate that both PIEZO1 and TRPV4 are required for normal mechanoelectrical transduction in murine chondrocytes in response to deflections applied at cell-substrate contact points. However, it is also clear that neither PIEZO1 nor TRPV4 are essential to this process, as deflection-gated currents were detected in *Trpv4*$^{-/-}$ cells and in chondrocytes treated with *Piezo1*-targeting miRNA. As such, we determined whether removal of both PIEZO1 and TRPV4 had an additive effect on chondrocyte mechanoelectrical transduction, using miRNA to knockdown *Piezo1* transcript in *Trpv4*$^{-/-}$ chondrocytes. In this case, significantly fewer cells (2/11) responded to deflection stimuli, compared with the WT chondrocytes treated with scrambled miRNA (Fisher's exact test, p=0.0002) (*Figure 4A*). The stimulus-response plot of *Trpv4*$^{-/-}$-*Piezo1*-KD chondrocytes was significantly different to that of scrambled miRNA-treated WT chondrocytes (Two-way ANOVA, p=0.04). In addition, the stimulus-response plot for *Trpv4*$^{-/-}$-*Piezo1*-KD cells highlights how little current activation was observed in the cells that responded to at least one stimulus (*Figure 4D*). These residual currents likely resulted from an incomplete knockdown of *Piezo1* transcript. We then asked whether these data reflect two subpopulations of cells, expressing either TRPV4 or PIEZO1, using calcium imaging experiments. Chondrocytes were loaded with the Cal520 calcium-sensitive dye and perfused with 10 μM ATP to test for viability. After ATP washout, cells were perfused with the PIEZO1 activator Yoda1 (10 μM). All the cells that had responded to ATP also exhibited an increase in $Ca^{2+}$ signal when treated with Yoda1. Following Yoda1 washout, the cells were then perfused with the TRPV4 agonist, GSK1016790A (50 nM). All the analyzed cells exhibited an increase in $Ca^{2+}$ signal when treated with GSK1016790A (400 cells, from two separate chondrocyte preparations; *Figure 4E*). These data clearly demonstrate that both PIEZO1 and TRPV4 are expressed and active in the membrane of all of the viable chondrocytes isolated from the articular cartilage.

## A TRPV4-specific antagonist, GSK205, reversibly blocks mechanically gated currents in chondrocytes

In order to definitively test whether TRPV4 is activated in response to substrate deflections, we used the TRPV4-specific antagonist GSK205 (*Vincent and Duncton, 2011*). We found that acute application of GSK205 (10 μM) reversibly blocked deflection-gated ion channel activity (n = 12 WT cells from five preparations) (*Figure 5A*). In the presence of GSK205, deflection-gated current amplitudes were significantly smaller, 13 ± 6% (mean ± s.e.m.) of pre-treatment values. After washout of the TRPV4 antagonist, current amplitudes recovered to 97 ± 28% of pretreatment values (*Figure 5B*) (one-way ANOVA, matched measures with Dunnett's post-hoc test for multiple comparisons. p=0.01

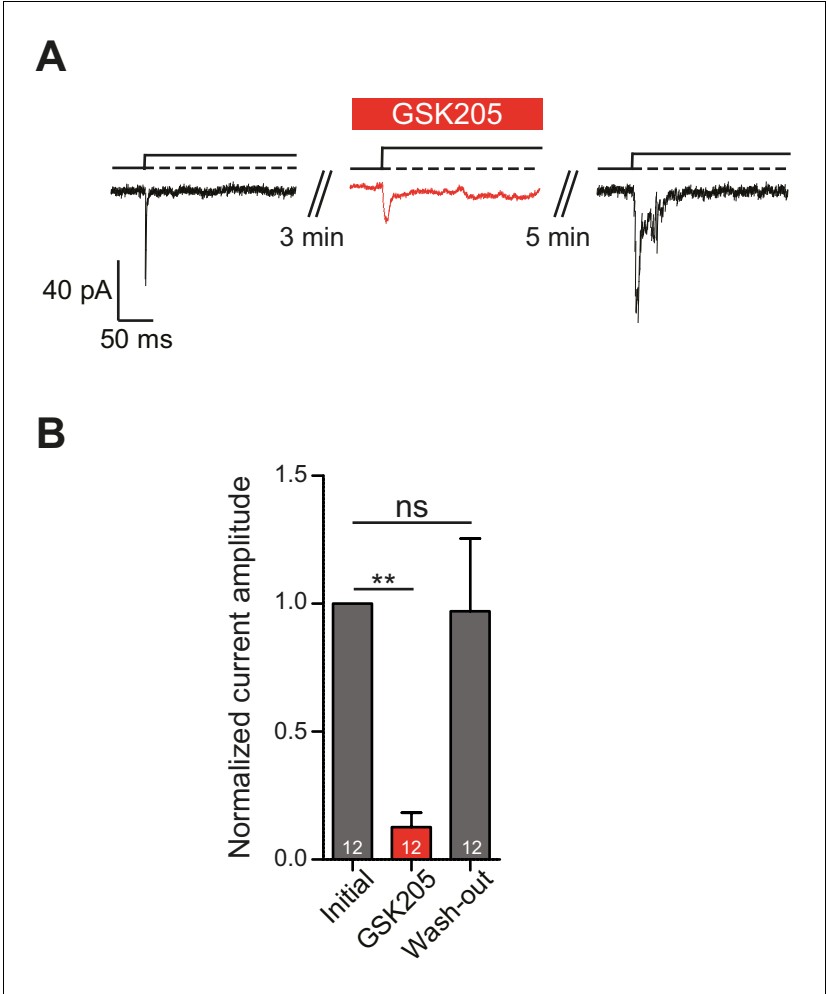

**Figure 5.** TRPV4 directly mediates deflection-gated currents in primary chondrocytes. (**A**) Representative traces of currents gated by pillar deflections before, during and after the wash out of the TRPV4 antagonist GSK205 (10 μM, 3 min). (**B**) Quantification of the inhibition of the current during the GSK205 application, the current amplitude was normalized against pre-treatment currents. Data represent average of 12 measurements. (One-way ANOVA, matched measures with Dunnett's post-hoc test for multibple comparisons. **p=0.01; ns = not significant).

treated vs pre-treated). These data indicate that TRPV4 directly mediates a large fraction of deflection-activated currents in WT chondrocytes.

## Stretch-activated channel activity in primary murine chondrocytes

Recently, evidence was provided from calcium-imaging experiments that TRPV4 in chondrocytes is activated by hypo-osmotic stimuli (*O'Conor et al., 2014*). Hypo-osmotic stimuli induce cell swelling (*Lechner et al., 2011*), and it has thus been postulated that TRPV4 is activated by the resulting membrane stretch. Accordingly, we investigated stretch-activated currents in outside-out membrane patches isolated from chondrocytes. We first tested chondrocytes transfected with either a scrambled miRNA or the *Piezo1*-targeting miRNA. It was not possible to generate a pressure-response curve using this second data set, as there was insufficient current activation over the range of applied stimuli. As such, we compared the peak current amplitude measured in outside-out patches. We found that when *Piezo1* was knocked down the stretch-mediated peak current amplitude measured using HSPC was 4.1 ± 0.8 pA (mean ± s.e.m., n = 10), significantly smaller than that measured in patches pulled from chondrocytes transfected with scrambled miRNA, 72.8 ± 14.3 pA (mean ± s.e.m., n = 11) (Student's *t*-test, p=0.0002) (*Figure 6A*).

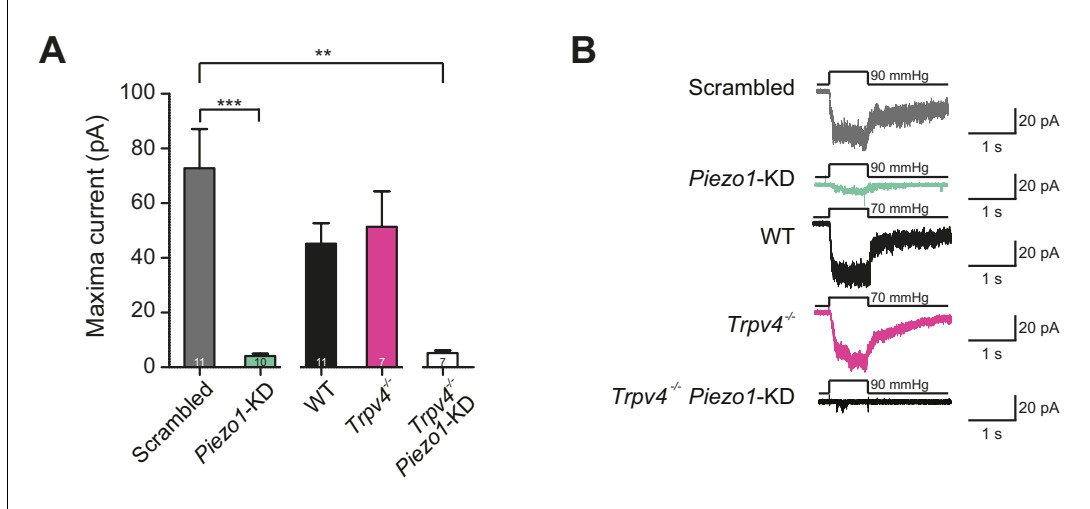

**Figure 6.** Murine chondrocytes display a stretch-sensitive current dependent on PIEZO1 but not TRPV4. (A) Comparison of maximal current induced by membrane stretch in outside-out patches isolated from chondrocytes. HSPC experiments were performed in membrane patches isolated from chondrocytes that were either: WT transfected with scrambled miRNA, WT transfected with *Piezo1*-targeting miRNA, WT, *Trpv4*-/- or *Trpv4*-/- transfected with *Piezo1*-targeting miRNA. WT chondrocytes transfected with *Piezo1*-targeting miRNA display significantly smaller maximal current amplitudes than WT chondrocytes transfected with scrambled miRNA (n = 11 and 10 patches, respectively, unpaired Student's *t*-test, ***p=0.0002). In contrast, peak current amplitude in *Trpv4*-/- chondrocytes was indistinguishable from that measured in WT chondrocytes. The treatment of *Trpv4*-/- chondrocytes with *Piezo1*-targeting miRNA led to a significant reduction in peak current amplitude compared to WT cells treated with scrambled miRNA (n = 7 and 11 patches, respectively, unpaired Student's *t*-test, **p=0.002). Number of *Trpv4*-/--*Piezo1*-KD chondrocytes: 11 scrambled-miRNA; 10 *Piezo1*-miRNA; 11 WT; 7 *Trpv4*-/-; 7 *Trpv4*-/-: *Piezo1*-miRNA. (B) Example traces of currents measured using HSPC in outside-out patches.

The following source data and figure supplements are available for figure 6:

**Source data 1.** Statistical comparison of stretch-gated mechanoelectrical transduction in chondrocytes.
**Figure supplement 1.** The $P_{50}$ measured in WT and *Trpv4*-/- chondrocytes using HSPC is not significantly different.
**Figure supplement 2.** WT chondrocytes respond to the TRPV4 agonist GSK101 but not chondrocytes isolated from a *Trpv4*-/- mouse.

We then compared outside-out patches isolated from WT chondrocytes to those isolated from *Trpv4*-/- mice. We found that patches pulled from WT chondrocytes exhibited robust currents to applied pressure, with a $P_{50}$ of 87.1 ± 6.0 mmHg (mean ± s.e.m., n = 12). However, we observed comparable stretch-activated currents in patches isolated from *Trpv4*-/- cells with a mean $P_{50}$ for activation (88.2 ± 9.3 mmHg (mean ± s.e.m., n = 7)) (**Figure 6—figure supplement 1**). In addition, there was no significant difference in peak current amplitude measured in these sample sets (*Trpv4*-/-, 51.4 ± 12.9 pA, n = 7; WT, 45.2 ± 7.5 pA, n = 12; mean ± s.e.m.) (**Figure 6A**). We confirmed that these cells lacked functional TRPV4 using the TRPV4-agonist GSK1016790A (**Figure 6—figure supplement 2**). When we treated *Trpv4*-/- cells with *Piezo1*-targeting miRNA we found that peak current amplitude (5.2 ± 0.9 pA, n = 7; mean ± s.e.m.) was significantly reduced, in comparison with the WT chondrocytes treated with scrambled miRNA (Student's *t*-test, p=0.002). The example traces presented in **Figure 6B** clearly demonstrate the loss of the stretch-activated current when *Piezo1* was knocked down.

These data demonstrate that PIEZO1 is largely responsible for the stretch-activated current in chondrocytes, whilst TRPV4 does not seem to play a role in this specific mechanoelectrical transduction pathway. In addition, the fact that stretch-activated currents in WT and *Trpv4*-/- cells were indistinguishable supports the hypothesis offered above that stretch-gated and deflection-gated currents represent distinct phenomena.

## In a heterologous system TRPV4 is gated efficiently by substrate deflections

TRPV4 is a polymodal channel (*Nilius et al., 2004*; *Darby et al., 2016*) that has been shown to be gated by diverse inputs, including temperature, osmotic and chemical stimuli (*Vriens et al., 2005*). In addition, TRPV4 has been demonstrated to play a role in mechanotransduction pathways in a variety of cells and tissues, including chondrocytes (*O'Conor et al., 2014*), vascular endothelium (*Thodeti et al., 2009*) and urothelium (*Miyamoto et al., 2014*; *Mochizuki et al., 2009*), yet it remains unclear whether TRPV4 is directly gated by mechanical stimuli or is activated down-stream of a force sensor (*Christensen and Corey, 2007*). In order to address this question, we asked whether the TRPV4 channel can be gated by various mechanical stimuli (applied using HSPC, cellular indentation or pillar deflection) when expressed in heterologous cells.

We first confirmed that we could measure robust PIEZO1-mediated currents in outside-out patches isolated from HEK-293 cells, where PIEZO1 was overexpressed. PIEZO1 exhibited large amplitude (>50 pA) and robust macroscopic currents in response to pressure-stimuli (*Figure 7B*, left panel). We also confirmed that PIEZO1 responds to indentation stimuli (*Figure 7B*, center panel), in accordance with published data (*Coste et al., 2012*; *Peyronnet et al., 2013*; *Gottlieb et al., 2012*; *Cox et al., 2016*). As shown previously (*Poole et al., 2014*) and confirmed here, PIEZO1 was also efficiently gated by deflection stimuli (*Figure 7B*, right panel). In previous studies, TRPV4 has been shown to respond to membrane-stretch when overexpressed in *X. laevis* oocytes (*Loukin et al., 2010*), but similar activity was not observed when TRPV4 was overexpressed in HEK-293 cells (*Strotmann et al., 2000*). We found that currents were observed in response to membrane-stretch but only in a subset of membrane patches (55%, 5/9 patches). Additionally, in those patches that did respond to pressure stimuli, we were unable to determine a $P_{50}$, as the currents putatively mediated by TRPV4 were not particularly robust (*Figure 7C*, left panel). In cell-free patches, TRPV4 is no longer activated by warm temperatures (*Watanabe et al., 2002*). These data indicate that outside-out patches lack functional molecular components necessary for some modes of TRPV4 activation. As such, we next tested whether TRPV4 was activated by stretch in cell-attached patches. Similar to the results obtained in outside-out patches, TRPV4 did not respond to stretch stimuli applied using HSPC (*Figure 7—figure supplement 1*). These data demonstrate that PIEZO1 is more efficiently gated by membrane-stretch than TRPV4, in a heterologous cell system.

We next tested whether cellular indentation could activate TRPV4 currents. We compared channel activity in HEK-293 cells measured using whole-cell patch-clamp in cells expressing PIEZO1, TRPV4 or LifeAct as a negative control. PIEZO1-mediated currents were measured in all cells (12 cells), in response to indentations of 0.5–11 µm, in accordance with published data (*Coste et al., 2012*; *Gottlieb et al., 2012*; *Coste et al., 2010*). In contrast, the response of HEK-293 cells expressing TRPV4 was indistinguishable from the negative control (*Figure 7C*, center panel; *Figure 7—figure supplement 2*).

TRPV4-expressing HEK-293 cells exhibited large currents in response to deflection stimuli in 87% transfected cells measured (39/45), in contrast to the lack of TRPV4 activation by pressure or indentation stimuli (*Figure 7C*, right panel). In order to confirm that the current observed in cells overexpressing TRPV4 was mediated by this channel, we acutely applied GSK205 (10 µM) and noted that with similar deflection stimuli the current was blocked. After wash-out of the TRPV4-specific antagonist, the amplitude of the mechanoelectrical transduction current was restored to pre-treatment levels (*Figure 8A*). These data clearly indicate that the deflection-gated current in HEK-293 cells overexpressing TRPV4 is mediated by the TRPV4 channel.

We compared the sensitivity of TRPV4 *versus* PIEZO1 and found that HEK-293 cells overexpressing TRPV4 exhibited larger currents in response to stimuli up to 500 nm, compared to HEK-293 cells overexpressing PIEZO1 (*Figure 8B*). The overall TRPV4 stimulus-response data were significantly different than for PIEZO1 (two-way ANOVA, p<0.0001). When we analyzed the current kinetics (*Figure 8C*), we found that both the activation time constant ($\tau_1$) and current decay ($\tau_2$) were significantly faster for TRPV4-mediated currents, compared to PIEZO1-mediated currents; however, there was no significant difference in the latencies for current activation. These data are pertinent because the latency and the activation time constant (namely, latencies below 5 ms and activation time constants faster than 1 ms) have been used as parameters to classify channels that may be directly gated by a mechanical stimulus, in this case a deflection stimulus. These rapid latencies (<5 ms) are distinct

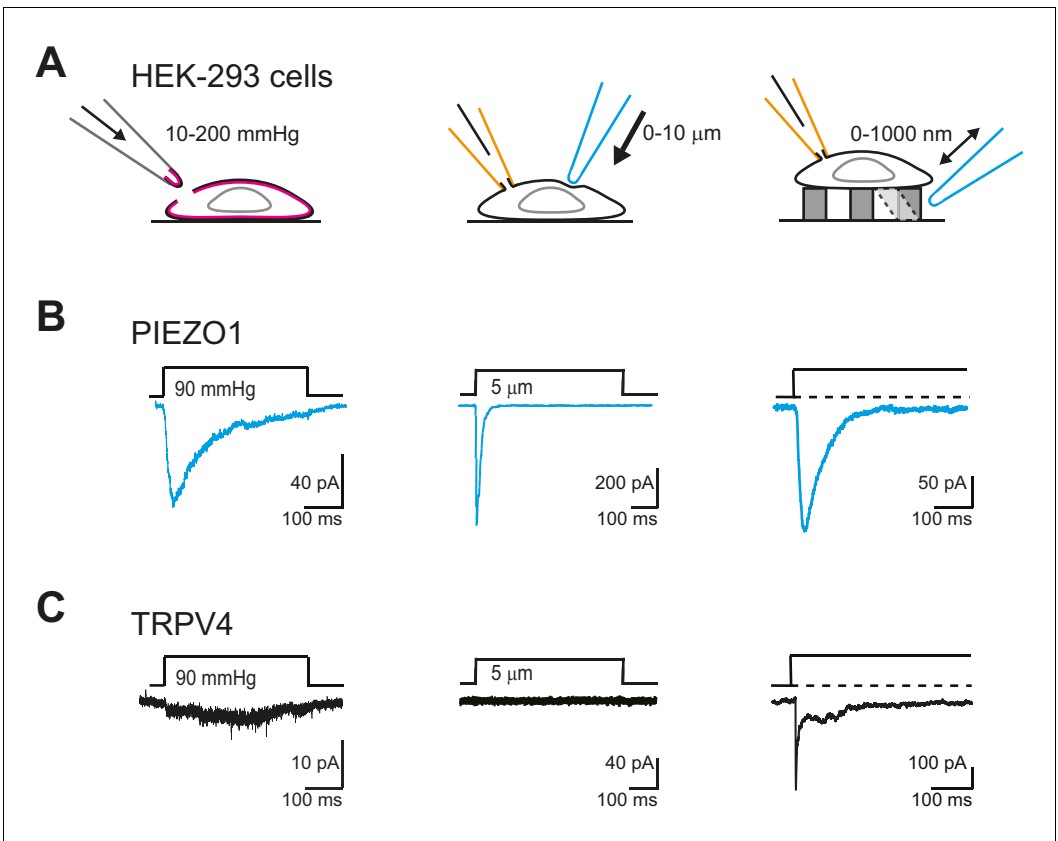

**Figure 7.** TRPV4 is efficiently gated by substrate deflections. (**A**) HEK-293 cells were used as a heterologous system to test stretch- indentation- and deflection-mediated currents. In the left panel is a cartoon of the HSPC experiment (stretch), in the center, indentation and on the right of the pillar array experiment (deflection). (**B**) PIEZO1 is efficiently gated by membrane stretch, indentation and substrate deflection. Left panel: example trace of PIEZO1-mediated current in an HSPC experiment, center panel: example trace from indentation experiment, right panel: example trace of PIEZO1-mediated current activated by substrate deflection. (cyan) (**C**) TRPV4 is efficiently gated by substrate deflection. Left panel: example trace (black) of HSPC of TRPV4 in outside-out patches isolated from HEK-293 cells. Center panel: example trace from HEK-293 cells expressing TRPV4 in response to indentation. Right panel: TRPV4 activation by substrate deflections in HEK-293 cells.

The following figure supplements are available for figure 7:

**Figure supplement 1.** Mechanical stimulation of cell-attached patches in HEK-293 cells overexpressing PIEZO1 or TRPV4.

**Figure supplement 2.** Mechanical indentation of HEK cells overexpressing PIEZO1 or TRPV4.

from the latencies of seconds to minutes measured for TRPV4 activation by osmotic stimuli, cell swelling, chemical activators (4αPDD) or heat (*Lechner et al., 2011*; *Nilius et al., 2004*, *2003*). We next measured the current amplitude of deflection-activated TRPV4 currents at negative and positive potentials (−60 mV versus 60 mV) to determine if these currents were outwardly rectifying, as observed for activation of TRPV4 by 4αPDD, swelling and heat (*Nilius et al., 2003*). The peak amplitude of the outward currents measured at 60 mV (157 ± 23 pA, mean ± s.e.m.; n = 30 currents) was not significantly different to the peak amplitude of the inward current measured at –60 mV (147 ± 15 pA, mean ± s.e.m.; n = 67 currents; Mann-Whitney U test). In addition, the current decay of the outward currents, measured at 60 mV (11.7 ± 4.8 ms, mean ± s.e.m.; n = 30 currents), was indistinguishable from inward currents measured at a holding potential of –60 mV (12.2 ± 2.7, mean ± s.e.m.; n = 67 currents; Mann-Whitney U test) (*Figure 8D*). These data stand in contrast to PIEZO1, where

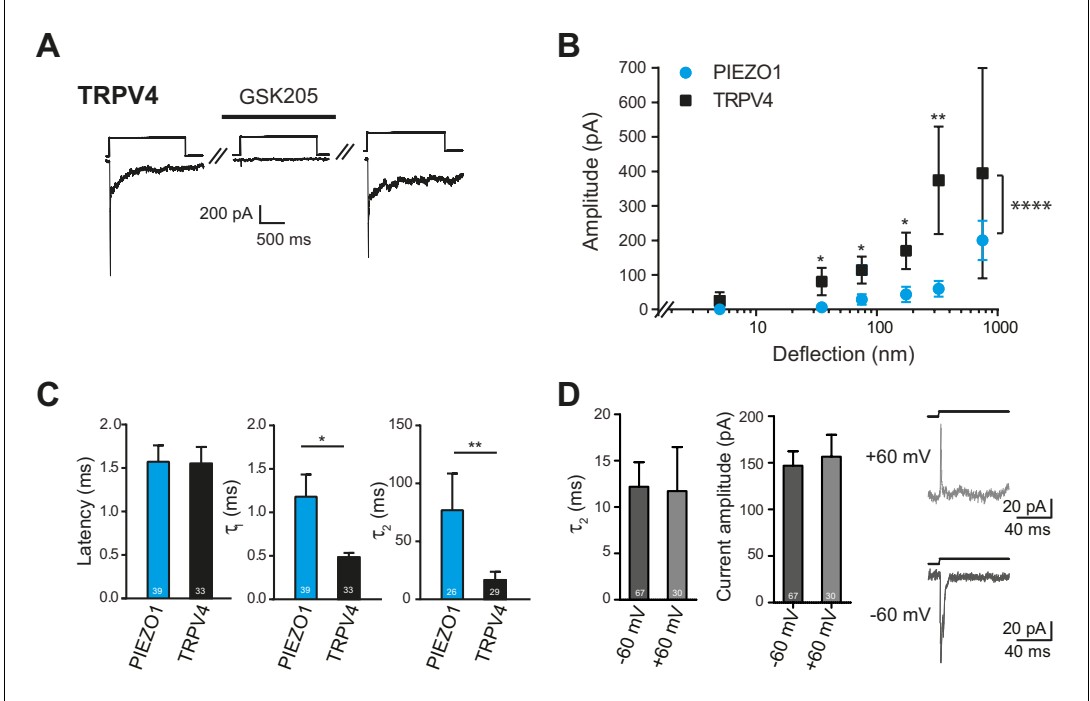

**Figure 8.** Deflection-mediated activation of TRPV4. (**A**) The deflection-gated current observed in HEK-293 cells expressing TRPV4 is reversibly blocked by the TRPV4-specific antagonist, GSK205 (10 μM, 3 min) (**B**) TRPV4 is more sensitive to substrate deflections than PIEZO1, in HEK-293 cells. Stimulus-response plots of current amplitude vs the magnitude of pillar deflection demonstrates that with stimulus sizes of 10–50, 50–100, 100–250, 250–500 nm cells expressing TRPV4 (black squares) (n = 8 cells) respond with significantly larger current amplitude than cells expressing PIEZO1 (cyan circles) (n = 12 cells) (see source data for details). In addition, the TRPV4 response is significantly different to the PIEZO1 response (two-way ANOVA, ****p<0.0001). (**C**) The kinetics of the deflection-gated currents. No differences were observed in the latency of current activation. However, both current activation ($\tau_1$) and current decay ($\tau_2$) values were significantly faster for TRPV4-mediated currents than PIEZO1-mediated currents. (Student's $t$-test; *p=0.04; **p=0.005). (**D**) Inactivation of TRPV4 at positive and negative potentials. Deflection stimuli were applied to HEK-293 cells expressing TRPV4 at – 60 mV and +60 mV. The current decay ($\tau_2$) and current amplitude values did not depend on the membrane holding potential (n = 67 currents, −60 mV; 30 currents, 60 mV; Mann-Whitney U test). Data collected from 16 cells over five experiments. Example traces are presented in the right hand panel.

The following source data and figure supplement are available for figure 8:

**Source data 1.** Electrophysiological characteristics of HEK-293 cells overexpressing either TRPV4 or PIEZO1.

**Figure supplement 1.** PLA2 is not involved in the activation by pillar-deflection of TRPV4

current decay values increased at positive membrane potentials (*Coste et al., 2012*, *2010*; *Gottlieb and Sachs, 2012*).

Activation of TRPV4 in response to cell swelling depends on phospholipase 2 (PLA2) enzymatic activity, releasing arachidonic acid from the membrane (*Vriens et al., 2004*). In order to determine whether such a second messenger system is required for deflection-mediated activation of TRPV4, we repeated the stimulus-response analysis in the presence of ACA, a PLA2 inhibitor. There were no significant changes in TRPV4-mediated currents in the presence of ACA, suggesting that gating of TRPV4 in response to cell swelling is distinguishable from the activation of TRPV4 in response to substrate deflections (*Figure 8—figure supplement 1*).

We conclude that TRPV4 directly mediates currents in response to deflection stimuli applied at cell-substrate contact points and that, in this transmembrane compartment, TRPV4 can be directly gated by the mechanical stimulus, as opposed to indirectly via a second messenger cascade. We propose that this mechanical activation of TRPV4 is distinct from the mechanisms of gating by heat, chemical activators and cell swelling.

## Discussion

We have addressed the questions of whether mechanically gated channel activity can be directly measured in primary murine chondrocytes, which channels mediate this process and how the specific type of mechanical stimulus affects mechanoelectrical transduction. In situ, chondrocytes are subjected to physical stimuli propagated via the fluid phase of the cartilage, as well as via contacts between the cells and ECM. Mechanical loading within the joints leads to chondrocyte deformations and changes in cell volume, applying strain to the cells in situ (*Guilak et al., 1995*; *Alexopoulos et al., 2005*; *Madden et al., 2013*). The transfer of mechanical loading to the chondrocytes themselves is modulated by the local mechanical environment, i.e. the local ECM structure and properties of the PCM (*Madden et al., 2013*). In vivo there exists a functional relationship between the PCM and the chondrocyte, together forming the chondron and changes in the composition or the mechanical properties of the PCM can lead to the development of OA (*Alexopoulos et al., 2009*; *Zelenski et al., 2015*). In this study, we have investigated mechanoelectrical transduction in isolated chondrocytes in response to deflections applied at the cell-substrate interface (to model stimuli transferred to the cells via matrix contacts) and to stretch applied to patches of membrane. We chose to directly monitor channel activity using electrophysiological techniques. Given that such an experimental approach requires access to the cell membrane, our studies have been conducted on chondrocytes in a 2D environment, as opposed to the 3D environment found in vivo.

Using pillar arrays, we were able to determine that the average substrate-deflection required for channel gating in chondrocytes was $252 \pm 68$ nm. Accordingly, chondrocyte mechanoelectrical transduction sensitivity to stimuli applied at the cell-substrate interface does not rival that of mechanoreceptor sensory neurons (known for their low mechanical threshold) but is comparable with the higher mechanoelectrical transduction threshold of nociceptive sensory neurons (*Poole et al., 2014*). Within the cartilage, chondrocytes are subjected to deformation but these shape changes are markedly different depending on the specific joint region (*Madden et al., 2013*; *Gao et al., 2015*). However, changes of 10–15% along the chondrocyte height axis in response to mechanical loading have been measured (*Amini et al., 2010*). Given that such changes represent average differences in cell length of >1 μm, this threshold lies within the range of conceivable membrane displacements that would occur in situ.

There is variation in the amplitude of the mechanically gated currents measured in response to pillar deflections, resulting in data with large error bars. We have noted this variability in all systems tested to date: sensory mechanoreceptive neurons, sensory nociceptive neurons, Neuro2A cells and HEK-293 cells heterologously expressing either PIEZO1 or PIEZO2. There are two likely reasons for this variability. Firstly, the pillar deflection stimuli are applied to a 10 μm$^2$ contact area between the cell and the pilus, restricting the number of potentially activated domains and resulting in noisier data than methods where stimuli are applied over a larger area, e.g. indentation. Secondly, stimuli are applied via dynamic cell-substrate contact points, likely introducing additional confounding factors such as changes in the local mechanical environment dictated by adhesion molecules and the cytoskeleton. It is interesting to note that, despite clear differences in mechanosensitivity between chondrocytes and dedifferentiated cells measured using pillar arrays, no differences were observed when HSPC was used to apply pressure-stimuli to membrane patches. This phenomenon may reflect differences in the mechanical environment of the cell matrix contact points in the spherical chondrocytes versus the flattened edges of the dedifferentiated cells that display a more fibroblast-like morphology. These data suggest that the behavior of mechanically gated channels in response to membrane stretch cannot be directly related to channel function when stimuli are applied via cell-substrate contact points and suggests that distinct pathways may mediate mechanoelectrical transduction within the cartilage in response to applied forces that stretch the membrane versus those forces propagated via movements within the matrix.

The elements of the pillar arrays are elastomeric cylinders, i.e. springs, meaning that the deflection of each pilus can be converted into a corresponding restoring force, using Hooke's Law (see Materials and methods). When we applied this conversion to our deflection data we obtained an average threshold for current activation of 63 nN in chondrocytes when deflection stimuli are applied to a 10 μm$^2$ patch of membrane, i.e. approximately 2% of the cell surface. These data do not indicate the force that is transferred to the mechanically gated ion channel, and this value for the restoring force will also be influenced by the mechanical properties of the cell at the cell-pilus contact.

However, given that the elasticity of chondrocytes (approx. 1 kPa (*Trickey et al., 2000*; *Shieh and Athanasiou, 2006*)) is three orders of magnitude lower than that of the substrate (2 MPa (*Poole et al., 2014*)), the influence of the mechanical properties of the cell on the restoring force will be minimal. These data allow a first comparison with earlier studies that investigated chondrocyte responses to compression. The calculated threshold for transduction in response to pillar deflection is almost 10x smaller than the compressive forces, applied to the whole cell, required in order to generate a robust $Ca^{2+}$ signal (500 nN, (*Lee, 2014*)). This comparison suggests that current activation is more sensitive to deflections applied at the cell-substrate interface than to whole-cell compression.

We have found that both TRPV4 and PIEZO1 are involved in mediating deflection-gated currents in chondrocytes. In the light of recent work on TRPV4 and PIEZO1 in porcine chondrocytes, it has been proposed that TRPV4 responds to fine mechanical stimuli and PIEZO1 to injurious stimuli (*Boettner et al., 2014*). In contrast, studies using $Ca^{2+}$ imaging to measure mechanotransduction in response to substrate-stretch in urothelial cells found that PIEZO1 mediates cellular mechanosensitivity in response to smaller stimuli than TRPV4 (*Miyamoto et al., 2014*). In both cases, the 'readout' of mechanotransduction is down-stream of the mechanoelectrical transduction event, monitoring alterations in matrix production (*O'Conor et al., 2014*) or changes in intracellular $Ca^{2+}$ levels (*O'Conor et al., 2014*; *Lee, 2014*; *Miyamoto et al., 2014*). As such, the relative differences in mechanosensitivity that depend on TRPV4 or PIEZO1 expression in the two systems could either reflect (a) differential modulation of channel sensitivity in distinct tissues by accessory molecules (as previously demonstrated for PIEZO1 [*Poole et al., 2014*]) or (b) that the pathways downstream of the channel event amplify the signal in a differential fashion. These two possibilities are also not mutually exclusive. Our data suggest that, in chondrocytes, it is the downstream amplification of the original mechanoelectrical transduction current that differs, as we observed very similar effects on mechanoelectrical transduction sensitivity when either TRPV4 or PIEZO1 levels were ablated. Some care does need to be taken with this interpretation due to the fact that a specific TRPV4-antagonist acutely and reversibly blocked 87% of the deflection-gated current, yet chondrocytes from $Trpv4^{-/-}$ mice did not display a similar reduction in current amplitude. We conclude that the chronic loss of one mechanosensitive channel in chondrocytes can be compensated for by other molecules, particularly given the fact that both TRPV4 and PIEZO1 were found to be active in all viable chondrocytes isolated from the articular cartilage. Such a conclusion supports the theory that there are multiple redundancies in mechanoelectrical transduction pathways (*Arnadóttir and Chalfie, 2010*) and highlights the possibility that potentially more mechanically gated channels await discovery.

Whilst both TRPV4 and PIEZO1 are required for normal mechanoelectrical transduction in response to substrate deflections, only PIEZO1 is required for normal current activation in HSPC measurements. A recent paper has demonstrated that PIEZO1 gating can be directly mediated by changes in membrane tension in membrane blebs (*Cox et al., 2016*), suggesting an underlying mechanism for this stretch-mediated channel gating. In our experiments, when *Piezo1* transcript levels in chondrocytes were knocked-down using miRNA, stretch-activated currents largely disappeared, whereas a complete absence of TRPV4 did not significantly change the peak current amplitude nor the $P_{50}$, in comparison with WT chondrocytes. This is a clear demonstration that current activation in response to membrane stretch cannot be used as an indicator of the overall mechanoelectrical transduction pathways within a cell. In addition, this observation highlights the impact of quantitative measurements of channel activity when precise stimuli are applied directly to a specific membrane environment, such as the cell-substrate interface.

Our data suggest that both PIEZO1 and TRPV4 similarly contribute to mechanoelectrical transduction of nanoscale deflection-stimuli in chondrocytes, whilst differing in their response to membrane stretch. We therefore addressed whether the two channels behave similarly in a heterologous system. We confirmed that TRPV4, unlike PIEZO1, is not efficiently gated by pressure-induced membrane-stretch, and demonstrated that TRPV4 is not activated by cellular indentation. It has previously been shown that TRPV4 can be gated by membrane-stretch in *X. laevis* oocytes (*Loukin et al., 2010*); however, the recording conditions used to demonstrate this effect all promote TRPV4 channel gating (holding potential + 50 mV, 20 mM Sodium Citrate and a pH of 4.5). Taken together with our observations, these data suggest that whilst TRPV4 can be gated by pressure stimuli, this process is not particularly efficient. However, we observed that HEK-293 cells expressing TRPV4 are more sensitive to mechanical stimuli applied at cell-substrate contact points than HEK-293 cells

expressing PIEZO1. For TRPV4-expressing cells, the latency between stimulus and response (<2 ms, indistinguishable from PIEZO1 expressing cells) and the activation time constant (<0.5 ms, significantly faster than PIEZO1-expressing cells) suggest that, in response to deflection stimuli, TRPV4 is directly gated by the mechanical stimulus. These data directly address the long-standing question of whether TRPV4 is a mechanically gated channel (*Christensen and Corey, 2007*). A number of criteria have been proposed to determine whether a channel is mechanically gated: the latency of current activation should be less than 5 ms (*Christensen and Corey, 2007*), the channel should be present in mechanosensitive cells, ablation of the channel should eliminate the response, expression of the channel in a heterologous system should produce mechanically gated currents and there should be an effect on mechanotransduction processes in vivo when the channel is deleted (*Arnadóttir and Chalfie, 2010*). As shown in this study, TRPV4-mediated current activation occurs with sufficiently rapid latencies. TRPV4 is expressed in the chondrocytes (along with other mechanosensory cells): its deletion leads to a reduction in mechanotransduction, in WT chondrocytes mechanotransduction currents are largely blocked by a TRPV4 antagonist and *Trpv4*$^{-/-}$ mice are more likely to develop OA (although given the polymodal nature of TRPV4 these changes do not definitively reflect changes in mechanoelectrical transduction). In addition, we demonstrate here that TRPV4 mediates mechanically-gated currents in response to substrate deflections in a heterologous system. Whilst the loss of this channel does not produce a complete loss of current, the observed redundancy in mechanoelectrical transduction pathways suggests that this criterion is too stringent.

We propose that studying how mechanically gated channels function when stimuli are applied at cell-substrate contact points will prove instrumental in elucidating the role of both TRPV4 and PIEZO1 in mechanosensing pathways in additional cell types. PIEZO1 has recently been shown to be inherently mechanosensitive (*Syeda et al., 2016*). In contrast, the data that we present here suggests that TRPV4 mechanosensitivity depends on the type of stimulus and the membrane compartment to which stimuli are applied. We speculate that differences in channel gating in response to physical stimuli applied to distinct membrane compartments represents a mechanism by which cells can promote mechanoelectrical transduction events to changes in the surrounding matrix without increasing cellular sensitivity to localized membrane stretch. As such, the direct measurement of mechanically gated ion channel activity in response to stimuli applied via cell-substrate contact points is essential in order to understand how cells respond to changes in their immediate physical environment.

## Materials and methods

### Molecular biology

The mouse-*TRPV4* in pcDNA3 plasmid was a kind gift from Dr. Veit Flockerzi (*Wissenbach et al., 2000*). For RT-qPCR experiments, total RNA was extracted using Trizol reagent (Ambion, Carlsand, CA, 15596018) according to manufacturer's instructions, contaminating genomic DNA was digested using the TURBO DNA-free kit (Ambion, AM1907) and 2 μg of RNA was reverse transcribed using random primers and SuperScript III (Invitrogen, Germany, 18080–044) according to manufacturer's instructions. The RT-qPCR reactions were performed in an Abi 7900 Sequence Detection System (Applied Biosystems, Germany) using probes from the Universal Probe Library Set (Roche, Germany, 04 683 641 001). Relative expression levels were calculated using the 2-ΔCt method with the house-keeping genes *β-actin* and *Hprt1*[75]. (Primer sequences, listed as 5' to 3': *β-actin*, AAGGCCAACCGTGAAAAGAT, GTGGTACGACCAGAGGCATAC; *Hprt1*, TCCTCCTCAGACCGCTTTT, CCTGGTTCATCATCGCTAATC; *Piezo1*, GACGCCTCACGAGGAAAG, GTCGTCATCATCGTCATCGT; *Piezo2*, ACGGTCCAGCTTCTCTTCAA, CTACTGTTCCGGGTGCTTG; *Trpv4*, CCACCCCAGTGACAACAAG, GGAGCTTTGGGGCTCTGT; *Sox9*, TATCTTCAAGGCGCTGCAA, TCGGTTTTGGGAGTGGTG.) When recovering cells from alginate beads due to the low number of cells, RNA was extracted using RNeasy microkit (Qiagen GmbH, Germany, 74004). miRNA sequences targeting *Piezo1* were previously cloned using the Block-iT Pol II miR RNAi system (Invitrogen) and validated along with a scrambled miRNA that does not target any known vertebrate gene was used as a control (*Poole et al., 2014*). The sequence of the selected miRNAs were: *Piezo1* targeting, top strand: 5'-TGCTGTAAAGATGTCCTTCAGGTCCAGTTTTGGCCACTGACTGACTGGACC

TGGGACATCTTTA-3', bottom strand: 5'-CCTGTAAAGATGTCCCAGGTCCAGTCAGTCAG TGGCCAAAACTGGACCTGAAGGACATCTTTAC-3'.

## Cell culture: Primary chondrocyte culture

Primary chondrocytes from mice (aged 4–5 days) were cultured as described previously (*Gosset et al., 2008*). Briefly, the knees and femoral heads were removed, mildly chopped and rinsed with PBS. The rinsed cartilage was treated with collagenase D (3 mg/ml, Roche 11 088 882001) in chondrocyte basal medium (Lonza, Walkersville, MD, CC-3217) for 1 hr. The cartilage was treated overnight with collagenase D (0.5 mg/ml) in medium with 10% FBS at 37°C. The suspension was then centrifuged at 400 $g$ for 10 min and the resulting pellet was incubated for 10 min with 0.05% trypsin-EDTA (Biotech GmbH, Germany, P10-023100) at 37°C. Chondrocytes were washed and harvested, then plated in flasks or encapsulated in alginate to maintain their differentiated state. Chondrocytes cultured in flasks were used only until passage 3. Mouse strains used in this study were WT C57Bl/6 from Charles River or $Trpv4^{-/-}$ (Jackson Laboratory, MGI ID: 2667379) on a C57Bl/6 background. To confirm a lack of functional TRPV4 in the $Trpv4^{-/-}$ mice each litter was genotyped using the suggested protocol from Jackson Laboratory and a sample of cells from each preparation was treated with the TRPV4 agonist GSK1016790A (Sigma Aldrich, G07898, 100 nM) and monitored using functional $Ca^{2+}$ analysis (*Figure 6—figure supplement 1*). Chondrocytes were transfected with Lipofectamine LTX and Plus Reagent (Invitrogen, 15338) according to manufacter´s instructions. All experiments involving mice were carried out in accordance with protocols approved by the German Federal authorities (State of Berlin).

## Encapsulation

Primary mouse chondrocytes were encapsulated in alginate (*Brand et al., 2012*). Briefly, chondrocyte density was adjusted to $8 \times 10^5$ cells/ml and cells were then mixed with 1 ml of alginate (1.2% w/v in solution: 25 mM HEPES, 118 NaCl, 5.6 KCl, 2.5 $MgCl_2$, pH 7.4) and passed dropwise through a 22-gauge needle into gelation solution (22 mM $CaCl_2$, 10 mM Hepes, pH 7.4). The encapsulated cells were cultured in Chondrocyte Differentiation Medium (Lonza, CC-3225). To recover the cells, the alginate matrix was dissolved using 55 mM Na citrate (Sigma Aldrich, Germany, 6132) at 37°C.

## Cell culture: cultured cell line

HEK-293 cells were used as a heterologous cell line to study TRPV4 and PIEZO1 activity. This cell line was chosen as it has previously been shown to exhibit little mechanoelectrical transduction in response to deflection stimuli within the 1–1000 nm range (*Poole et al., 2014*). HEK-293 cells were cultured in DMEM media containing 10% fetal calf serum and 1% penicillin, streptomycin. To transfect HEK-293 cells, FuGeneHD (Promega, Madison, WI, E231A) was used as per manufacturer's instructions. HEK-293 cells were tested regularly to confirm absence of mycoplasma, using a luminescence kit from Epo GmBH (Germany), as per manufacturer's instructions. The identity of the cultured cells was authenticated by Eurofins Medigenomix Forensik GmbH (Germany), using PCR-single-locus-technology using 21 independent PCR reactions.

## Inmunofluorescence

For immunofluorescence staining, cells were fixed with 4% PFA. When labeling intracellular components, cells were permeabilized with 0.25% Triton-X 100 (Sigma-Aldrich, X-100). Fetal goat serum (3%) in PBS was used as a blocking agent before labeling with primary antibody (anti-Sox9 (Abcam, UK, ab59265; at 1:500), anti-Collagen X (Abcam, ab49945; at 1:2000)). Secondary antibodies were all used at a dilution of 1:2000 (Life Technologies, Germany, A11034, A31630, A21050).

## Preparation of pillar arrays

Pillar arrays were prepared as described previously (*Poole et al., 2014*); briefly, a silanized negative master was coated with degassed PDMS mixed at a ratio of 10:1. After 30 min, the still-liquid PDMS was covered with a glass coverslip (thickness, 2) and the coated master placed at 110°C for 1 hr. After curing, the pillar array was gently peeled away from the master. Before use, pillar arrays were either coated with PLL or activated by plasma cleaning (Deiner Electronic GmbH, Germany) and cells were allowed to attach.

The individual elements in the pillar arrays were cylinders made of PDMS, as such, they could be modeled as a spring. Therefore, deflection measured (d) could be converted into a corresponding force using Hooke's law (*Equation 1*),

$$F = -kd,$$  (1)

where $k$ is the spring constant of each pilus in the array. The spring constant is dependent on the elasticity (E) of the material, and the dimensions of the cylinder, according to *Equation 2*:

$$k = \frac{3}{4} \cdot \pi \cdot E \cdot \frac{r^4}{L^3}$$  (2)

The arrays were cast under curing conditions that result in an elasticity of the PDMS equal to 2.1 MPa and the dimensions of the elements within the array were: radius = 1.79 μm; length = 5.87 μm (*Poole et al., 2014*). The spring constant of each individual pilus was therefore 251 pN/nm.

To generate quantitative data on mechanoelectrical transduction an individual pilus subjacent to a cell was deflected using a polished glass probe (approx. 2 μm in diameter) driven by a MM3A micromanipulator (Kleindiek Nanotechnik, Germany). The electrical response of the cells was monitored using whole-cell patch-clamp and to quantitate the magnitude of the stimulus, a bright-field image was taken before pillar deflection, during the applied stimulus and after the release of the stimulus. Bright-field images were taken using a 40x objective and a CoolSnapEZ camera (Photometrics, Tucson, AZ). To calculate the pillar deflection, the center point of the relevant pilus was determined from a 2D Gaussian fit of the intensity values in the relevant images (Igor, Wavemetrics, TIgard, OR); the distance that this center point moves represents the stimulus magnitude. The estimated error of the calculated stimulus size was 7 nm, as previously described (*Poole et al., 2014*).

## Electrophysiology

Whole-cell patch-clamp recordings were performed at room temperature. The resistance of the recording pipettes ranged between 3 and 5 MΩ. Currents were acquired at 10 kHz and filtered at 3 kHz using an EPC-10 amplifier with Patchmaster software (HEKA, Elektronik GmbH, Germany) in combination with a Zeiss 200 inverted microscope and were analyzed using FitMaster software (HEKA, Elektronik GmbH). The bath solution contained (in mM) 140 NaCl, 4 KCl, 2 CaCl$_2$, 1 MgCl$_2$, 4 glucose and 10 HEPES, adjusted to pH 7.4 with NaOH. The internal solution contained (in mM) 110 KCl, 10 NaCl, 1 MgCl$_2$, 1 EGTA and 10 HEPES, adjusted to pH 7.3 with KOH. The membrane potential was held at −40 mV in chondrocyte measurements (*Sánchez and Wilkins, 2003*; *Sánchez et al., 2006*) and −60 mV for HEK-293 cell measurements. GSK205 (Calbiochem, Billerica, MA, 616522) was used at a concentration of 10 μM and cells were treated for 3 min. ACA (Calbiochem, 104550) was used at a concentration of 20 μM and applied directly via the patch pipette. We allowed solution exchange for at least 3 min before collecting data.

## Cellular indentation

Cellular indentation studies were performed as described previously (*Hu and Lewin, 2006*). Briefly, cells were indented using a fire-polished glass probe, with a diameter of approximately 2 μm. The probe was moved toward the cell until compression of the surface was observed. Indentation stimuli (between 0.5–11 μm) were then applied by driving the glass probe into the cell using the MM3A micromanipulator. Cellular responses were simultaneously monitored using whole-cell patch-clamp.

## High-speed pressure clamp

Outside-out patches were pulled from cells and currents were elicited by applying positive pressure to the patch via the patch pipette using a High Speed Pressure Clamp (ALA Scientific, Farmingdale, NY). Within 30 s of pulling the patch a protocol of pressure steps (duration 600 ms, application 0.1 Hz) ranging from 10 mmHg to 150 mmHg, in 20 mmHg steps were applied while holding the patch at −60 mV. The sensitivity of stretch-activated channels for each patch was estimated by fitting individual pressure response curves to the Boltzmann equation. Extracellular solution had the following composition (in mM): 150 NaCl, 5 KCl, 10 Hepes, 10 glucose, 1 MgCl$_2$, 2 CaCl$_2$. The intracellular solution contained (in mM): 140 KCl, 10 Hepes, 1 EGTA, 1 MgCl$_2$. Thick

walled electrodes (Harvard apparatus 1.17 mm x 0.87 mm, external and internal diameter respectively) were pulled with a DMZ puller (Germany) polished to a final resistance of 6 to 8 MΩ. Cell-attached recordings were performed using the extracellular solution in the pipette and at a holding voltage of −60 mV. Negative pressure steps were applied with the same frequency amplitude as for outside-out patches.

### Calcium imaging

Chondrocytes were plated on PLL-coated glass coverslips and loaded with Cal-520 (5 µM) for 1 hr (AAT-Bioquest, Sunnyvale, CA). Cells were placed in 200 µl of solution in a chamber that allows laminar flow and a fast solution exchange. Calcium images were acquired using a DG4 (Sutter Instruments, Novato, CA) as a light source and were acquired and analyzed using Metafluor (Molecular Devices, Sunnyvale, CA). Fluorescent images were acquired every 5 s. The initial (background) fluorescence was acquired for 10 cycles and used to normalize the fluorescence of the whole experiment. Fluorescence values were calculated and plotted according to the formula $DF/F = (F-F_0)/(F_0)$ where $F_0$ is baseline fluorescence for Cal-520. Yoda1 (10 µM) was applied for 90s, followed by wash-out period of 5 min, whereas GSK1016790A (50 nM) was applied for 15 s.

### Statistical analysis

The stimulus-response data collected from experiments performed in the pillar arrays have variation in $x$ (deflection) and $y$ (current amplitude); therefore, the response was grouped in bins of different sizes in order to compare it. The size of the bins is as follows: 0–10, 10–50, 100–250, 250–500 and 500–1000 nm. For each cell, the current amplitudes within the bins are averaged, and then these data averaged across cells. All data sets were tested for normality: parametric data sets were compared using a two-tailed, Student's $t$-test, paired or unpaired depending on the experimental setup, nonparametric data sets were compared using a Mann-Whitney test. In order to compare the overall response of samples to deflection stimuli, we conducted two-way ANOVA. Categorical data were compared using Fisher's exact test. One-way ANOVA and Tukey post-hoc test were used to compare RT-qPCR data sets.

## Acknowledgements

The authors would like to thank Raluca Fleischer, Liana Kosizki, Anke Scheer and Karola Bach for technical assistance and Jeffrey H. Stear for critical reading of the manuscript. This work was supported, by a Cecile Vogt Fellowship from the MDC to KP, a Collaborative Research Center Grant, from the Deutsche Forschungsgemeinschaft, SFB958 (Project A09, to KP and GRL) and a GoldStar Award from the University of New South Wales to KP.

## Additional information

### Funding

| Funder | Grant reference number | Author |
| --- | --- | --- |
| Deutsche Forschungsgemeinschaft | SFB958 (A09) | Gary R Lewin Kate Poole |
| Max Delbruck Center for Molecular Medicine | Cecile Vogt Fellowship | Kate Poole |
| University of New South Wales | GoldStar Award | Kate Poole |

The funders had no role in study design, data collection and interpretation, or the decision to submit the work for publication.

### Author contributions

MRS-V, KP, Conception and design, Acquisition of data, Analysis and interpretation of data, Drafting or revising the article; MM, Acquisition of data, Analysis and interpretation of data; GRL, Conception and design, Drafting or revising the article

Author ORCIDs
Gary R Lewin, http://orcid.org/0000-0002-2890-6352
Kate Poole, http://orcid.org/0000-0003-0879-6093

Ethics
Animal experimentation: All experiments involving primary tissue isolated from mice were carried out in accordance with protocols approved by the German Federal authorities (State of Berlin), under the specific license number X9014/15.

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
