## [Decision Letter]

Thank you for submitting your article "Direct measurement of TRPV4 and PIEZO1 activity reveals multiple mechanotransduction pathways in chondrocytes" for consideration by *eLife*. Your article has been reviewed by three peer reviewers, one of whom, Kenton Swartz, is a member of our Board of Reviewing Editors, and the evaluation has been overseen by Anna Akhmanova as the Senior Editor.

The reviewers have discussed the reviews with one another and the Reviewing Editor has drafted this decision to help you prepare a revised submission.

Summary:

In the skeletal system, cartilage that covers bone joints, and chondrocytes present within cartilage, is exposed to extensive mechanical changes. Response of chondrocytes to mechanical stimulation has been well characterized but the mechanotransduction channel or the molecular mechanism involved in sensing biophysical changes in this environment remains elusive.

There is evidence in the literature implicating TrpV4 in chondrocyte mechanotransduction but none conclusively demonstrating that they are mechanically activated ion channels. For example, mutations in Trpv4 have been implicated in skeletal dysplasia and joint dysfunction, and mice lacking TrpV4 are easily susceptible to osteoarthritis, a condition caused due to disrupted extracellular matrix homeostasis in chondrocytes. However, TRPV4 is a polymodal sensor activated by a variety of stimuli, and nailing down a role of TRPV4 mechanotransduction in vivo has been a challenge. On the other hand, Piezos are bona fide mechanically activated ion channels, but their role as the primary mechanotransducer in murine chondrocyte biophysics hasn't been explored yet.

In the current manuscript Servin-Vences et al. address the role of Piezo1 and TrpV4 in chondrocyte mechanotransduction. As a first, they report mechanically activated currents recorded from primary murine chondrocytes, and show that Piezo1 and TrpV4 together mediate these currents. In primary chondrocytes and heterologous expression in HEK cells, Piezo1 can be mechanically activated by membrane stretch as well as stimulation of the cell-substrate interface, while TrpV4 is activated only when the cell-substrate is mechanically stimulated and not by direct membrane stretch. Thus suggesting that TrpV4 mechanical activation occurs only when the cell is mechanically stimulated via the extracellular matrix.

The data presented in the manuscript support the idea that Piezo1 and TrpV4 mediate the mechanotransduction signal in chondrocytes. However, it is important to reiterate that TRPV4 is a polymodal sensor, and whether TRPV4 is playing a role in mechanotransduction in vivo has not been demonstrated yet. Therefore, we think it is important for the authors to discuss this issue in the Discussion section. Overall, we support the publication of this manuscript but would like the authors to address the following issues:

Essential revisions:

1) The authors claim that selective activation of TrpV4 by deflecting the cell-substrate interface is observed because the pillar deflection assay is more sensitive than apical membrane indentation or membrane stretch (both of which are extensively characterized and widely used and accepted mechanical stimulation assays in the field of mechanotransduction). An alternative, and more relevant, conclusion is that TrpV4 is activated only when the cell membrane is stretched via the extracellular matrix and not by direct membrane stretch. One way to test this is to check if mechanical indentation of the apical membrane results in TrpV4 activation. If one doesn't see TrpV4 mediated currents then it only means that TrpV4 mechanical activation is dependent on the cell matrix, which could also explain why TrpV4 KO mice are more prone to osteoarthritis which is a result of disrupted ECM homeostasis. Beyond proving any mechanism, testing mechanical indentation for TrpV4 activation is important, as it is a very commonly used method for mechanotransduction, and it is important to compare to responses to deflections (as is already done with stretch).

2) An important conclusion of the work is that stretch (in HSPC experiments) and "substrate deflections" reflect different mechanical stimuli that activate distinct mechanoelectrical transduction pathways. This is mainly based on the finding that TRPV4-dependent currents are activated by moving a pilus, but not in the HSPC experiments using outside-out patches. However, it is well known that in cell-free patches components that are critical for TRPV4 function are lost, which may well explain the absence (or rather, the small amplitude) of TRPV4-dependent currents in the HSPC experiments. For instance, it is known that TRPV4 loses heat sensitivity in cell-free patches (see Chung et al. JBC, 2003; Watanabe et al. JBC, 2002), so it is conceivable that TRPV4 can be activated by stretch, but that the responsiveness is lost upon patch excision. Thus, additional experiments comparing how Piezo and TRPV4 respond to membrane stretch in cell-attached patches would be necessary if the authors want to make the case that activation of TRPV4 in the pillar deflection experiments is not due to membrane stretch.

3) The authors calculate a force from the displacement of the pilus, which they use to determine a "threshold for current activation" (subsection “Chondrocytes and dedifferentiated cells display distinct mechanosensitivity”). This calculation, based on Hooke's law, is in our opinion incorrect, as it only represents the force needed to deflect the pilus "in isolation". However, given that the pilus displacement activates a current in the cell, there must be a force applied from the pilus on to the cell and from the cell on the pilus. These forces depend on factors that are unknown, including the strength of the interaction between membrane and pilus (i.e. is the pilus tightly attached to the membrane patch, or rather rubs it), and the mechanoelastical properties of the cells, and thus invalidate the force estimations. In simple words, if the cell would be made out of concrete, the force to create a certain pilus displacement would be much higher than if the cell were made of butter. Thus, any conclusion on absolute force or differences in force threshold seems inappropriate at this point.

4) Much of what is described here for the Piezo1-currents is similar to what is already described in several earlier papers from these and other authors. The real novelty lies in the direct mechanical activation of TRPV4. However, it would be ideal if this novel aspect of the work could be extended. For instance, the mechanosensitive current attributed to TRPV4 shows very rapid inactivation, which has not been described for this channel for any other stimulus, but is partly reminiscent of the properties of Piezo's. Description of some basic properties of the mechano-activated TRPV4-currents (e.g. voltage dependence of the inactivation, recovery from inactivation, selectivity etc.) would add to the novelty aspect of the manuscript.

5) In Figure 4 the authors show that KD of Piezo1 or TrpV4 results in only 50% of the tested cells responding to mechanical stimulation. While, the effect is additive when both Piezo1 and TrpV4 is knocked down simultaneously. This suggests that TrpV4 and Piezo1 are expressed in different cells. However, from Figure 4 it appears that there is a reduction in currents in the remaining cells when either Piezo1 or TRPV4 is targeted. This is confusing. The first set of experiments suggest that Piezo1 and TRPV4 are expressed in mainly non-overlapping cells. If this was the case, one would not expect reduced current in other half when each is individually targeted. The authors should evaluate TRPV4 and Pieoz1 expression in individual chondrocytes and attempt to make sense of the recordings.

6) Overall, the relevance of the applied stimuli for chondrocyte function in vivo is quite vague and not well discussed. Neither the HSPC nor the pillar structure seem to match well with the structures and forces that are experienced by these cells in situ. The relevance of comparing chondrocytes with the dedifferentiated cells is also not clear. The authors should discuss these issues openly in the manuscript.

7) Figure 5. It is puzzling that the application of the TrpV4 antagonist GSK205 results in almost 80% block of the MA current, which we know is a combination of Piezo1 and TrpV4. This suggests that GSK205 inhibits Piezo1 as well? The authors should test the effect of the antagonist on Piezo1 to see if this is true. Also, an N of 4 in this experiment is low, given the importance it holds. Finally, it appears that the current was rescued in only 3/4 cells tested?

8) The error bars for most of the pillar deflection experiments are large, indicating variability. Does the peak current vary so much between cells? And, this is seen both in primary chondrocytes and HEK cells. Can the authors comment on what would cause this?

---

## [Author Response]

*Essential revisions:*

*1) The authors claim that selective activation of TrpV4 by deflecting the cell-substrate interface is observed because the pillar deflection assay is more sensitive than apical membrane indentation or membrane stretch (both of which are extensively characterized and widely used and accepted mechanical stimulation assays in the field of mechanotransduction). An alternative, and more relevant, conclusion is that TrpV4 is activated only when the cell membrane is stretched via the extracellular matrix and not by direct membrane stretch. One way to test this is to check if mechanical indentation of the apical membrane results in TrpV4 activation. If one doesn't see TrpV4 mediated currents then it only means that TrpV4 mechanical activation is dependent on the cell matrix, which could also explain why TrpV4 KO mice are more prone to osteoarthritis which is a result of disrupted ECM homeostasis. Beyond proving any mechanism, testing mechanical indentation for TrpV4 activation is important, as it is a very commonly used method for mechanotransduction, and it is important to compare to responses to deflections (as is already done with stretch).*

We have conducted indentation experiments using HEK-293 cells:

– overexpressing PIEZO1 or

– overexpressing TRPV4 or

– transfected with LifeAct-mCherry as a negative control.

In accordance with previously published data, we can measure currents activated by indentation when PIEZO1 is overexpressed. In contrast, we do not measure robust channel gating when cells expressing TRPV4 are indented, and these data are indistinguishable from the negative controls. These data have been added to a reconfigured Figure 7 (PIEZO1 and TRPV4 traces). We have also prepared a figure supplement containing further current traces including those acquired from the negative control. We found this to be important as on occasion, small, slowly inactivating currents are measured in the negative control cells in response to large stimuli. In the interests of transparency we have provided these data in the new Figure 7—figure supplement 2.

We have also modified the text to reflect that these data strengthen the conclusion that the TRPV4 channel activation depends on the application of mechanical stimuli via the matrix, as suggested by the reviewers above.

Our contention that the use of pillar arrays is a more sensitive approach stems from our earlier work investigating mechanoelectrical transduction in dorsal root ganglia neurons. In that system we could measure currents in response to pillar defection or indentation. However, the deflection of the membrane required to activate channels in the pillar array experiment was significantly smaller than the size of the indentation required for robust channel gating. We have now clarified in the text to avoid confusion.

The addition of these data means that we have now compared our pillar array measurements with both commonly used methods for studying mechanical gating of ion channels, i.e. pressure clamp and indentation.

*2) An important conclusion of the work is that stretch (in HSPC experiments) and "substrate deflections" reflect different mechanical stimuli that activate distinct mechanoelectrical transduction pathways. This is mainly based on the finding that TRPV4-dependent currents are activated by moving a pilus, but not in the HSPC experiments using outside-out patches. However, it is well known that in cell-free patches components that are critical for TRPV4 function are lost, which may well explain the absence (or rather, the small amplitude) of TRPV4-dependent currents in the HSPC experiments. For instance, it is known that TRPV4 loses heat sensitivity in cell-free patches (see Chung et al. JBC, 2003; Watanabe et al. JBC, 2002), so it is conceivable that TRPV4 can be activated by stretch, but that the responsiveness is lost upon patch excision. Thus, additional experiments comparing how Piezo and TRPV4 respond to membrane stretch in cell-attached patches would be necessary if the authors want to make the case that activation of TRPV4 in the pillar deflection experiments is not due to membrane stretch.*

We have now conducted experiments in HEK-293 cells overexpressing either PIEZO1 or TRPV4 using HSPC analysis of cell-attached patches.

– Consistent with data published previously by other groups we can measure robust mechanically-activated PIEZO1 currents.

– In contrast we do not measure any appreciable TRPV4-mediated currents.

These data have been included as Figure 7—figure supplement 1 and the appropriate Watanabe, JCB 2002 reference added to the reference list.

We feel that this was an excellent suggestion from the reviewers and that these data, together with the indentation results only strengthen our conclusions.

*3) The authors calculate a force from the displacement of the pilus, which they use to determine a "threshold for current activation" (subsection “Chondrocytes and dedifferentiated cells display distinct mechanosensitivity”). This calculation, based on Hooke's law, is in our opinion incorrect, as it only represents the force needed to deflect the pilus "in isolation". However, given that the pilus displacement activates a current in the cell, there must be a force applied from the pilus on to the cell and from the cell on the pilus. These forces depend on factors that are unknown, including the strength of the interaction between membrane and pilus (i.e. is the pilus tightly attached to the membrane patch, or rather rubs it), and the mechanoelastical properties of the cells, and thus invalidate the force estimations. In simple words, if the cell would be made out of concrete, the force to create a certain pilus displacement would be much higher than if the cell were made of butter. Thus, any conclusion on absolute force or differences in force threshold seems inappropriate at this point.*

We agree with the reviewers that there are significant limitations in interpreting these data as a force. Indeed, we write in the original manuscript “These data do not indicate the force that is transferred to the mechanically-gated ion channel” and we have presented our stimulus magnitude as a deflection in all of the figures because of these concerns. We have now updated this section of the Discussion to note that the physical properties of the cell will also confound this force calculation. However, the deformability of chondrocytes has been empirically tested and values are reported in the literature of around 1-2 kPa. Given that the modulus of the substrate that we use is 2.1 MPa the cell is 3 orders of magnitude more compliant than the substrate. So whilst there would be an influence on the force calculation due to the physical interaction with the cell, we feel there is still value in reporting the restoring force in the discussion due to the fact that the properties of the substrate will predominate.

Our motivation for reporting a force was to allow us to compare the values that we have measured with our system with those collected using the cantilever of an Atomic Force Microscope to compress the cells.

Given that this force measurement is problematic we have removed the section reporting the threshold in pN force from the results, leaving all reporting of thresholds in the Results section in nm deflection.

We have maintained the section in the Discussion, adding additional caveats to make clear that care should be taken in drawing conclusions from these values. We feel it is important to include this conversion in the discussion (with the appropriate cautions) to allow some comparison across methodologies. That is, to allow some degree of comparison with the compression data obtained using atomic force microscopy.

*4) Much of what is described here for the Piezo1-currents is similar to what is already described in several earlier papers from these and other authors. The real novelty lies in the direct mechanical activation of TRPV4. However, it would be ideal if this novel aspect of the work could be extended. For instance, the mechanosensitive current attributed to TRPV4 shows very rapid inactivation, which has not been described for this channel for any other stimulus, but is partly reminiscent of the properties of Piezo's. Description of some basic properties of the mechano-activated TRPV4-currents (e.g. voltage dependence of the inactivation, recovery from inactivation, selectivity etc.) would add to the novelty aspect of the manuscript.*

We have now added data to the new Figure 8 showing that there is no voltage-dependence of the inactivation of the current. In addition, we found that this TRPV4-mediated current does not seem to be outwardly rectifying. This is in contrast to studies on activation of TRPV4 by 4αPDD, heat and cell swelling. Taken with the rapid latency of activation, we speculate that these data reflect a distinct pathway of TRPV4 activation, as previously reported for swelling vs heat.

*5) In Figure 4 the authors show that KD of Piezo1 or TrpV4 results in only 50% of the tested cells responding to mechanical stimulation. While, the effect is additive when both Piezo1 and TrpV4 is knocked down simultaneously. This suggests that TrpV4 and Piezo1 are expressed in different cells. However, from Figure 4 it appears that there is a reduction in currents in the remaining cells when either Piezo1 or TRPV4 is targeted. This is confusing. The first set of experiments suggest that Piezo1 and TRPV4 are expressed in mainly non-overlapping cells. If this was the case, one would not expect reduced current in other half when each is individually targeted. The authors should evaluate TRPV4 and Pieoz1 expression in individual chondrocytes and attempt to make sense of the recordings.*

In our original manuscript we propose that PIEZO1 and TRPV4 both mediate currents activated by substrate deflections across all of the chondrocytes BUT that in the absence of one channel, the second can partially compensate.

As such, when TRPV4 is acutely blocked using GSK205 we measure an 85% inhibition, but when cells are isolated from a *Trpv4^-/-^*mouse PIEZO1 can somewhat compensate, meaning only 50% cells are affected.

In contrast, we propose that PIEZO1 is essential for the current that we observe in response to membrane stretch applied using HSPC:

– All of the cells treated with miRNA targeting Piezo1 exhibited smaller responses to membrane stretch

– Stretch-activated currents in chondrocytes isolated from the *Trpv4^-/-^* mouse are indistinguishable from those in WT cells.

In response to the reviewers’ questions regarding these ambiguities we tested whether both channels are present and active in all WT chondrocytes or whether there are sub-populations of cells with distinct channel expression profiles.

– We stimulated cells using pharmacological tools and observed cellular responses using fluorometric [Ca^2+^] imaging.

– We chose this approach over studying expression in individual cells as these data indicate whether or not functional channels are present within the plasma membrane.

– Application of either Yoda1 (PIEZO1 activator) or GSK1016790A (TRPV4 agonist) resulted in an increase in the calcium signal in all cells that had initially responded to ATP.

These data clearly demonstrate that both channels are present and functional in all of the chondrocytes, rather than segregated into separate sub-populations. We have added these data to Figure 4. We feel that this has strengthened this section of the manuscript and allowed a more informative interpretation of our results.

We contend that the differences between the two experiments reflect the difference between an acute and a chronic manipulation

– acutely blocking TRPV4 represents the contribution of TRPV4 in the WT cells vs

– chronic absence of the TRPV4 channel is partially compensated for by PIEZO1

We have clarified this in the text in the light of the new [Ca^2+^] data.

*6) Overall, the relevance of the applied stimuli for chondrocyte function* in vivo *is quite vague and not well discussed. Neither the HSPC nor the pillar structure seem to match well with the structures and forces that are experienced by these cells in situ. The relevance of comparing chondrocytes with the dedifferentiated cells is also not clear. The authors should discuss these issues openly in the manuscript.*

In the Introduction to our original manuscript we addressed the complex biomechanical environment of chondrocytes in articular cartilage. We have now added additional text to the Discussion to address how our experiments may relate to cellular function within this in vivo environment.

It is indeed difficult to model the complex physical impacts experienced by chondrocytes and here we have made the first steps to address how stretching the membrane alone compares to deflecting regions of cell-substrate contacts. The use of electrophysiology to directly monitor channel activity means that cells must be accessed with a patch pipette. As such, we are working with cells in a 2D environment rather than 3D. Clearly we are addressing a very localised phenomena, yet we contend that this is an important first step to highlight;

– the fact that the local channel environment can modulate channel activation.

– That there are distinct but overlapping mechanoelectrical transduction pathways reflecting the fact that TRPV4 only responds to mechanical stimuli propagated via cell-matrix contacts

Our data additionally suggest a mechanistic pathway for how changes in the ECM composition and compliance may lead to osteoarthritis.

With respect to why we compared chondrocytes and dedifferentiated cells: this question was motivated by logistics and current available published data.

– Logistically, it is much easier to culture, manipulate with molecular biology and patch dedifferentiated cells. If these cells had proven at the start of our study to model chondrocyte mechanoelectrical transduction we would have made all subsequent measures on such dedifferentiated cells.

– Given the noted differences we concluded that such an experimental approach was not viable.

– We have included these data in our manuscript as much of the published data relating to chondrocytes has been collected from chondrocytes that have been in 2D culture for more than 24 hours; such cells would be predominantly dedifferentiated.

As such, we found it necessary to include this information in our manuscript. We have adjusted the text in the Results sections to make clear our motivation for the comparison with regards to defining our sample set.

*7) Figure 5. It is puzzling that the application of the TrpV4 antagonist GSK205 results in almost 80% block of the MA current, which we know is a combination of Piezo1 and TrpV4. This suggests that GSK205 inhibits Piezo1 as well? The authors should test the effect of the antagonist on Piezo1 to see if this is true. Also, an N of 4 in this experiment is low, given the importance it holds. Finally, it appears that the current was rescued in only 3/4 cells tested?*

We have tripled our data set, with data now collected from 12 cells.

–The missing final current was our error, and has now been added.

–With the increased numbers the current amplitude measured in the presence of GSK205 were only 13 ± 6% of the pretreatment values, returning to 97 ± 28% on washout. A one-way ANOVA (matched measures) with Dunnett’s post-hoc tests indicated that the current amplitude was significantly different in the presence of GSK205 compared to pre-treatment current amplitudes, and the washout values were significantly different to those in the presence of GSK205. However, current amplitude after washout was not significantly different to the current amplitudes measured pre-treatment. These updated data have been added to the manuscript and Figure 5 has also been updated.

GSK205 is reported to be a TRPV4 specific antagonist. Given that the reviewers raise the question of specificity here and later in this review we have empirically tested whether GSK205 also inhibits PIEZO1-mediated currents. We have not included these data in the manuscript as GSK205 is reported throughout the literature to be a TRPV4-specific antagonist but we provide the data here for the reviewers.

– We performed indentation experiments on Neuro2A cells expressing PIEZO1, in the presence and absence of GSK205.

– Indentation stimuli were applied before, during and after the application of GSK205 (10 µM, 3 min; washout 5 min). Current amplitudes were normalised against the pre-treatment current.

– We found that normalised currents increased in the presence of GSK205 and increased further on washout. These data were indistinguishable from the response of cells perfused with extracellular solution instead of GSK205.

– As such, we conclude that GSK205 is not an antagonist of PIEZO1, at the concentration used in our manuscript.

Author response image 1.(**A**) Indentation of Neuro2A cells. Neuro2A cells were indented using a glass probe and resulting mechanically-gated currents were monitored using whole-cell patch-clamp. Indentation stimuli were applied before, during and after application of GSK205 (10 µM, 3 min; washout 5 min). Current amplitudes were normalised against pre-treatment currents. (**B**) Indentation experiments were repeated on cells treated with Extracellular solution instead of GSK205. Data represents average ± s.e.m., n = 6 cells (GSK205) vs 5 cells (negative control), from two separate experiments.**DOI:**
http://dx.doi.org/10.7554/eLife.21074.023

8) The error bars for most of the pillar deflection experiments are large, indicating variability. Does the peak current vary so much between cells? And, this is seen both in primary chondrocytes and HEK cells. Can the authors comment on what would cause this?

The error bars are indeed large. This is seen across all of our experiments using pillar arrays, including in our previously published research (Poole et al. 2014). We have noted this variability in all systems tested: sensory mechanoreceptive neurons, sensory nociceptive neurons, Neuro2A cells, HEK-293 cells heterologously expressing PIEZO1 or PIEZO2 and of course here chondrocytes, dedifferentiated cells and HEK-293 cells expressing TRPV4.

We contend that there are likely two fundamental reasons for the observed variation:

1) That by restricting the area to which we apply stimuli we are restricting the number of domains that can be stimulated, thus our data are intrinsically more noisy and,

2) Given that the interactions between the cell and the substrate are dynamic there are additional confounding factors that increase the variability in the response to stimulation. These confounding factors likely include differences in the precise molecular and mechanical environment leading to variations in the transfer of force from the pilus to the membrane.

We have added text to the Discussion to comment on this property of the pillar array experiments.